# Learning-based multi-objective hyper-heuristic algorithm for reconfigurable assembly line scheduling problems

**Haoyi Zhao**[1,2,3], **Xiangming Huang**[1], **Guoliang Liu**[3], **Zixiang Li**[4]*, **Fan Chen**[2], **Gaojie Lu**[2]

**1** College of Mechanical and Vehicle Engineering, Hunan University, Changsha, Hunan, China, **2** Wuhan Vocational College of Software and Engineering (Wuhan Open University), Wuhan, Hubei, China, **3** Hunan Sinoboom Intelligent Equipment Co., ltd., Changsha, Hunan, China, **4** Key Laboratory of Metallurgical Equipment and Control Technology of Ministry of Education, Wuhan University of Science and Technology, Wuhan, Hubei, China

* lizixiang@wust.edu.cn

## Abstract

Reconfigurable assembly lines have emerged as a vital manufacturing paradigm to meet the growing demand for customized and multi-variety products. This study considers the reconfigurable assembly line scheduling problem, involving product sequencing optimization, to minimize reconfiguration cost, production workload equalization, and logistics leveling simultaneously. This study formulates a novel and linearized multi-objective mathematical model, which rectifies deficiencies in prior formulations. A novel Q-learning-based multi-objective hyper-heuristic algorithm is proposed. The algorithm integrates multiple metaheuristic operators, including particle swarm optimization, teaching–learning-based optimization, whale optimization algorithm, and grey wolf optimizer, within a unified search framework. Q-learning is employed to adaptively select the most promising operator at each search stage based on real-time performance feedback. Moreover, the proposed algorithm incorporates a new density-aware leader selection strategy with a survival-time decay factor to select the global best solution for population evolution, favoring superior solutions in sparse regions and increasing selection pressure on high-quality individuals. A numerical case study demonstrates that the models with the ε-constraint method could achieve a set of Pareto solutions. A computational study on 120 generated benchmark instances demonstrates that the proposed methodology outperforms nine other high-performing multi-objective algorithms.

## 1. Introduction

With the development of customization, there is an increasing demand for producing multi-variety and small-lot products in the modern market. The modern manufacturing industry is developing new and flexible manufacturing systems to replace the

**Data availability statement:** All data, code, and computational results of this study are accessible on GitHub at https://github.com/zixiangliwust/Instances_RALSP under the MIT license.

**Funding:** This project is partially supported by National Natural Science Foundation of China under grant 62173260 and Hubei Provincial Natural Science Foundation of China under grant 2026AFD061.

**Competing interests:** The authors have declared that no competing interests exist.

traditional production models [1]. The reconfigurable manufacturing system (RMS), introduced by Koren, Heisel [2], is conceived to achieve high flexibility. RMS consists of modular machines that could modify production capacity and functionality by adding or subtracting hardware components and modifying software. And hence, RMS has a rapid response to fluctuations in market demand [3,4]. Compared to other systems, RMS has the advantage of swiftly switching between different products, and it could facilitate small-lot production and meet demands for multi-variety and customized output. RMS could effectively combine the high throughput rates of traditional assembly lines and the high flexibility of cellular systems [5]. For producing a single-type product, the basic simple-model assembly line suffices [6,7]. When it is necessary to assemble multiple products, there are two options: the mixed-model assembly line and the multi-model assembly line. Mixed-model assembly lines could assemble different products simultaneously on the same line. And the products on this line are usually structurally similar, and they could be represented by a joint precedence graph [8,9]. In this mixed-model assembly line, the reconfiguration costs and times for switching between products are usually negligible. Conversely, if product variations are significant, multi-model assembly lines are employed. Here, products are assembled in separate batches, requiring physical reconfiguration of the line between batches, and thus incurring non-negligible reconfiguration costs and time. Unlike mixed-model lines, multi-model lines assemble only one product type at any given time.

Fig 1 illustrates three assembly line configurations under different production demands. In this figure, a single-model assembly line could produce one single-type product (represented by triangles). Mixed-model assembly lines could assemble several products simultaneously, where different geometries denote different products. In contrast, the multi-model assembly line produces different products in separate batches, and the physical reconfiguration between batches is required (indicated by "R" in the diagram). For this configuration, the different sequences of products could lead to different reconfiguration costs. And the sequences of products also influence production workload balance and logistics leveling. Consequently, the optimization of product sequencing is essential for achieving high efficiency and low cost of reconfigurable assembly lines.

Despite several studies having been published, there are several gaps related to the reconfigurable assembly line scheduling problem (RALSP). Firstly, published mathematical models often suffer from nonlinearities and logical inconsistencies; the formulated formulations cannot be solved by standard optimization tools. Secondly, while some metaheuristics have been applied, there is a lack of intelligent, adaptive hyper-heuristic frameworks capable of dynamically leveraging the strengths of multiple search operators specifically for the multi-objective RALSP. Thirdly, the integration of advanced machine learning techniques like Q-learning for adaptive operator selection within such a framework remains underexplored. In this study, a more realistic RALSP is formulated with the objectives of minimizing reconfiguration cost, production workload equalization, and logistics leveling. A Q-learning-based multi-objective

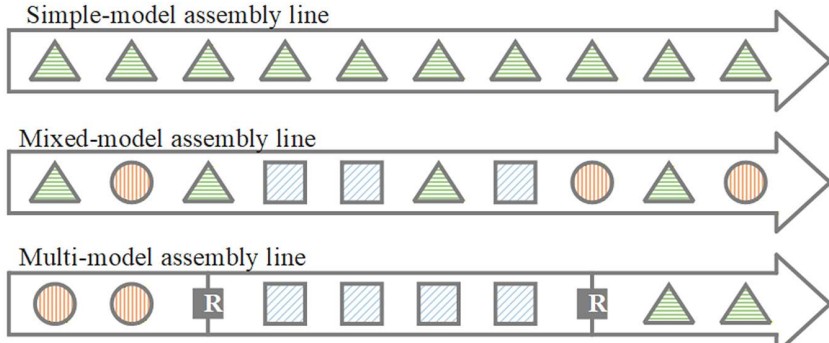

**Fig 1. Three assembly line configurations under three production demands.**

hyper-heuristic algorithm (QMOHH) is developed to achieve a set of Pareto solutions. The main contributions of this paper are as follows.

1) A novel and tractable multi-objective mathematical model is constructed to minimize reconfiguration cost, production workload equalization, and logistics leveling. This model advances beyond prior work by presenting three fully linearized mixed integer linear programming (MILP) sub-models. It rectifies errors found in previously published formulations and ensures direct solvability for benchmark generation and exact analysis. A numerical case study demonstrates that the models with the ε-constraint method could achieve a set of Pareto solutions.

2) The proposed QMOHH functions as a Q-learning-based hyper-heuristic framework that dynamically orchestrates multiple metaheuristic operators based on four metaheuristics—particle swarm optimization (PSO), teaching–learning-based optimization (TLBO), whale optimization algorithm (WOA), and grey wolf optimizer (GWO). Meanwhile, it employs a new density-aware leader selection strategy with a survival-time decay factor to select the global best solution for population evolution. This design ensures that the search maintains an effective equilibrium between thorough exploration and intensive exploitation.

3) A comprehensive comparative study is conducted to evaluate the performance of the proposed QMOHH. The algorithm is compared with nine other state-of-the-art multi-objective algorithms, including generalized differential evolution algorithm (GDE3), evolutionary algorithm based on decomposition (MOEAD), non-dominated sorting genetic algorithm II (NSGA-II), and others. Using 120 generated benchmark instances and three performance indicators, the experimental results show that the QMOHH consistently outperforms the competing algorithms.

The remainder of this paper is organized as follows. Section 2 reviews the relevant literature. Section 3 describes the problem and provides the mathematical formulation. Section 4 introduces the proposed QMOHH in detail. Section 5 provides a numerical case study to clarify the main features of the problem under consideration. Section 6 evaluates the improvements of the proposed QMOHH and provides a comparative study. Finally, Section 7 concludes this study and suggests future research directions.

## 2. Literature review

This section reviews relevant literature on RALSP, hyper-heuristic methods, and Q-learning integration. And then it concludes by identifying the research gaps and contributions.

## 2.1 Previous work on RALSP

Research on the RALSP focuses on flexibility improvement, multi-objective optimization, and adaptation to dynamic environments. Among pioneering works, Koren, Heisel [2] defined the RMS paradigm using modular machine tools and open-architecture controllers. After that, researchers have expanded the research on RMS in various dimensions. Specifically, Hasan, Jain [4] proposed a "Service Level" index to quantify reconfiguration effort. Colledani, Gyulai [10] developed an integrated method for system design and reconfiguration planning under uncertain demand. Dou, Li [11] adopted the NSGA-II for bi-objective optimization of cost and tardiness in reconfigurable flow lines.

For multi-objective optimization of RALSP, Goyal and Jain [12] combined MOPSO with maximum deviation theory. Yuan, Deng [13] considered cloud manufacturing and used distance-sorting PSO. And Prasad and Jayswal [14] incorporated reconfiguration effort, profit-to-cost ratio, and due dates using Shannon entropy. For modular products and dynamic environments, Pattanaik and Jena [15] used a Pareto-based heuristic. Yuan, Yu [16] designed a memetic algorithm. And Yang, Liu [17] used hybrid PSO for assembly sequence and equipment selection. Meanwhile, some attention is paid to reconfiguration efficiency and sustainability. Specifically, Yelles-Chaouche, Gurevsky [18] formulated MILP for task reassignment cost. Tremblet, Yelles-Chaouche [19] considered uncertain product arrival. Delorme and Gianessi [20] minimized power peaks. In addition, Gholami, Delorme [21] used fuzzy logic for sustainable supply chains.

As the application of hyper-heuristics to RALSP is limited, this section reviews the integration of hyper-heuristics and reinforcement learning in the related scheduling problems. Cano-Belmán, Ríos-Mercado [22] developed a scatter search-based hyper-heuristic for mixed-model sequencing. Mosadegh, Fatemi Ghomi [23] developed Q-learning-based simulated annealing. Afterwards, Özbakır and Seçme [24] proposed a hyper-heuristic for stochastic parallel lines. And Zhou and Zhao [25] developed a hyper-heuristic to optimize the material feeding. Guo, Liu [26] proposed the hyper-heuristic for integrated process planning and scheduling. Lu, Gao [27] solved PCB assembly line scheduling problems with a hyper-heuristic optimizer. Reviews on reinforcement learning-based hyper-heuristics and scheduling applications can be found in Li, Wei [28] and Vela, Valencia-Rivera [29].

Apart from the integration of hyper-heuristics and reinforcement learning, the integration of reinforcement learning (especially Q-learning) with metaheuristic algorithms has produced promising performance through adaptive scheduling. For instance, Yan and Wang [30] proposed a double-layer Q-learning algorithm in aircraft assembly scheduling, and Zhang, Tang [31] developed a Q-learning-based multi-objective evolutionary algorithm in assembly line balancing. In recent years, Meng, Li [32] proposed a Q-learning-inspired differential evolution algorithm for mixed-model assembly line balancing and sequencing. Rauf, Mumtaz [33] developed a multi-objective intelligent hybrid genetic algorithm integrated with Q-learning-based parametric tuning, and Wen, Liu [34] developed PSO with Q-learning for aircraft pulsating assembly line scheduling. For broader reviews, see Bortolini, Galizia [5] on RMS, Gianassi, Leoni [1] on resource management in mixed-model and multi-model assembly lines, and Battaïa, Delorme [35] on line balancing and model sequencing.

## 2.2 Research gap and contributions

Through the literature review, several research gaps can be identified. Firstly, the existing mathematical models for RALSP might have the following drawbacks: nonlinearities and logical inconsistencies. They are usually developed for describing the problem, and they cannot be solved by the standard optimization tools. Secondly, while some metaheuristics have been developed, there lacks the studies on the intelligent and adaptive hyper-heuristic frameworks, which could dynamically leverage the strengths of multiple metaheuristics for a special multi-objective RALSP. Thirdly, the integration of machine learning (e.g., Q-learning) into metaheuristic or hyper-heuristics for adaptive operator and parameter selection still needs more attention. To address these gaps, the main contributions of this paper are as follows.

1) **Formulation of a novel and tractable multi-objective mathematical model:** This study develops three fully linearized MILP sub-models—one for each objective: reconfiguration cost, workload equalization, and logistics leveling. This

formulation corrects flaws in earlier models and is directly solvable by commercial solvers, providing exact benchmarks for small to medium instances.

2) **Development of one new and effective hyper-heuristic:** The proposed QMOHH selects one promising metaheuristic operator from eight metaheuristic operators with the Q-learning. Meanwhile, it utilizes a density-aware leader selection strategy with a survival-time decay factor to select the global best solution for population evolution. The density-aware leader selection strategy provides more computation effort to the isolated Pareto solutions and ensures that each solution will be explored in iterations, hence ensuring an effective balance between exploration and exploitation.

3) **Comprehensive experimental study and validation:** This study conducts a comprehensive evaluation of the QMOHH utilizing 120 generated benchmark instances. And QMOHH is compared with nine other popular multi-objective algorithms. Computational results along with statistical analysis demonstrate the superiority of the proposed QMOHH in convergence, diversity, and overall solution quality.

The proposed QMOHH differs from existing reinforcement-learning-assisted metaheuristics in three key aspects. 1) Most studies embed Q-learning into a single metaheuristic (e.g., Mosadegh, Fatemi Ghomi [23]; Zhang, Tang [31]). QMOHH instead acts as a hyper-heuristic that selects among eight complete operators from four metaheuristics (PSO, TLBO, WOA, GWO), leveraging diverse search behaviors within a unified framework. 2) Existing approaches often use algorithm-specific states (e.g., diversity, temperature) and scalar fitness rewards. QMOHH defines the state purely by search progress and the reward as the number of newly added non-dominated solutions, directly aligning adaptation with multi-objective optimization goals. 3) Unlike crowding-distance-based or random selection, QMOHH introduces a survival-time decay factor that reduces the attractiveness of long-surviving solutions, preventing stagnation and promoting exploration in sparse regions.

## 3. Problem description and model formulations

This section starts with describing the problem under consideration in detail, where the characteristics and the assumptions are provided. Afterwards, this section presents the complete mathematical formulation, which consists of three interconnected sub-models.

### 3.1 Problem description

In practice, assembly lines are primarily categorized into three types, as illustrated in Fig 2. Single-model assembly lines are dedicated to the high-volume production of a single product type, where no sequencing problem arises due to fixed product variety, and the primary optimization objective is line balancing—assigning assembly tasks to stations based on one or several criteria. Mixed-model assembly lines assemble a family of products that are similar in structure, function, and assembly methods to meet multi-variety, small-batch demands, requiring simultaneous consideration of both line balancing and product sequencing. Reconfigurable assembly lines, designed for customized and small-batch production, sequence a set of structurally similar but distinct products by rapidly switching between them via hardware and software adjustments. In most studies, line balancing in a reconfigurable assembly line is assumed to be predetermined, so optimization focuses on product sequencing with objectives of minimizing reconfiguration cost, equalizing production workload, and leveling material logistics.

The RALSP addressed in this study involves sequencing $nz$ product types from the same family. The demand for the $n$th product type is $D_n (n = 1, 2, \cdots, nz)$, and the total demand is $D(D = \sum_{n=1}^{nz} D_n)$. The number of stations and the processing time for each product at each station are predetermined and fixed. To simplify the problem, a minimum production cycle is often used in the literature [13]. Here, the original demand $D_n$ is replaced by $d_n = \frac{D_n}{g}$, where $g$ is the greatest common divisor of the demands $D_n$ for all $nz$ product types. Consequently, the number of products to be sequenced is reduced to $d \left(d = \sum_{n=1}^{nz} d_n\right)$. For example, if the demands for four products (A, B, C, D) are 200, 200, 300, and 100, respectively, and their greatest common divisor is 100, the minimum production cycle is calculated as [1–3].

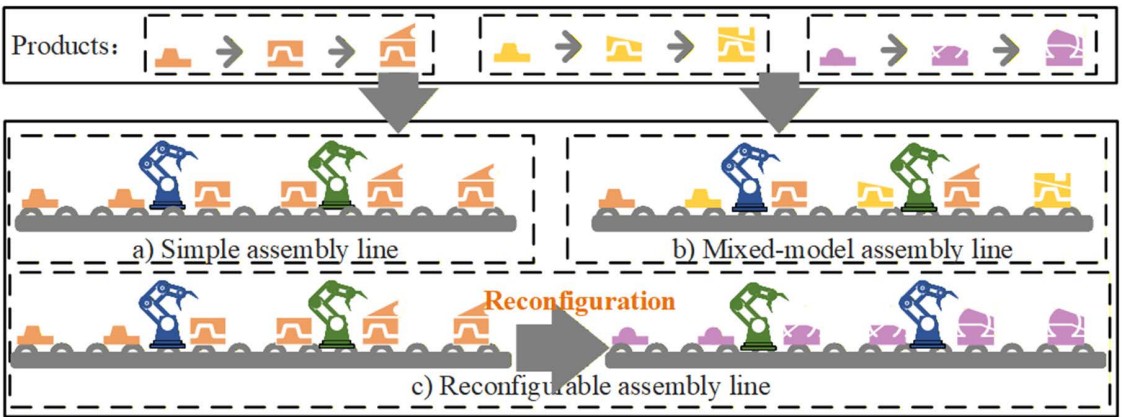

**Fig 2. Production scheduling characteristics of the three assembly line types.**

Before presenting the mathematical formulation, the following basic assumptions are presented: 1) the number of product types and stations is fixed and known; 2) the reconfiguration cost for switching between any two product types is known in advance; 3) the part requirements for each product type and the associated part-frequency constraints are known; 4) the processing time for each product type at each station and the constant interval time between successive product startups are known; 5) all stations are utilized for assembling every product, with the assembly process for each starting at the first station and finishing at the last; 6) and the transportation time for moving products between stations is negligible.

### 3.2 Mathematical model

This section presents the complete and tractable mathematical formulation for the RALSP. A key contribution of this study is the development of three fully linearized MILP models, each dedicated to one of the three conflicting objectives: reconfiguration cost minimization, production workload equalization, and logistics leveling. Unlike previous studies that often present conceptual or nonlinear formulations that are challenging to solve directly, the developed models are designed for solvability by standard MILP solvers (e.g., CPLEX, Gurobi), providing exact benchmarks and enabling practical deployment for small-to-medium instances. The parameters and variables are first described below.

| Symbol | Description | Dimensions |
|---|---|---|
| Sets and Indices | | |
| $M$ | Set of product types, $M = \{1, 2, \ldots, nm\}$. | |
| $K$ | Set of stations, $K = \{1, 2, \ldots, nk\}$. | |
| $R$ | Set of parts, $R = \{1, 2, \ldots, nr\}$. | |
| $D$ | Set of product positions, $D = \{1, 2, \ldots, nd\}$. | |
| $i, j$ | Product type indices, $i, j \in M$. | |
| $k$ | Station index, $k \in K$. | |
| $p$ | Position in product sequence, $p \in D$. | |
| $l$ | Part index, $l \in R$. | |
| $b$ | Index within consecutive positions, $b \in \{1, 2, \ldots, B_l\}$. | |

| Symbol | Description | Dimensions |
|---|---|---|
| Parameters | | |
| $nm$ | Number of product types. | |
| $nk$ | Number of stations. | |
| $nr$ | Number of parts. | |
| $nd$ | Number of products to be sequenced, $nd = \sum_{i \in M} d_i$. | |
| $d_i$ | Quantity of product $i$ in the minimum production cycle. | $nm$ |
| $C_{i,j}$ | Reconfiguration cost from product $i$ to product $j$. | $nm \times nm$ |
| $z_{i,l}$ | 1, if product $i$ requires part $l$; 0, otherwise. | $nm \times nr$ |
| $A_l, B_l$ | Part-frequency constraints; in any $B_l$ consecutive positions, at most $A_l$ products requiring part $l$. | $nr$ |
| $PT_{i,k}$ | Processing time of product $i$ at station $k$. | $nm \times nk$ |
| $IT$ | Inter-station interval time (constant). | |
| $BigM$ | Large positive constant (big-M parameter). | |

The first model (Model 1) addresses the objective of minimizing the total reconfiguration cost incurred when switching between different product types on the assembly line. In multi-model assembly lines, reconfiguration costs are high and cannot be neglected. Reconfiguration costs consist of expenses caused by changing hardware components, changing tools, reprogramming software, and downtime during changeovers. The minimization of reconfiguration cost is crucial to maintain the production efficiency and the profit of the assembly line. The decision variables for Model 1 are first introduced as follows.

| Variable | Domain | Description |
|---|---|---|
| $x_{p,i}$ | $\{0, 1\}$ | 1, if the product at position $p$ is of type $i$; 0, otherwise. |
| $y_{p,i,j}$ | $\{0, 1\}$ | 1, if the product at position $p$ is of type $i$ and the product at position $p + 1$ is of type $j$; 0, otherwise. |

Model 1 minimizes total reconfiguration cost. Equation (1) sums the costs of transitions between consecutive products, including from the last to the first. Constraints (2) and (3) enforce that each position has exactly one product and that demand is satisfied. Constraints (4)–(6) linearize the product-of-binary terms for consecutive positions using auxiliary variables $y_{p,i,j}$; constraints (7)–(9) handle the cyclic transition. The complete linearization is further discussed in Section 3.3.

$$\text{Min } F_1 = \sum_{p=1}^{nd} \sum_{i \in M} \sum_{j \in M} y_{p,i,j} \cdot C_{i,j} \tag{1}$$

$$\sum_{i \in M} x_{p,i} = 1 \quad \forall p \in D \tag{2}$$

$$\sum_{p \in D} x_{p,i} = d_i \quad \forall i \in M \tag{3}$$

$$y_{p,i,j} \leq x_{p,i} \quad \forall p \in \{1, \ldots, nd-1\}, \forall i, j \in M \tag{4}$$

$$y_{p,i,j} \leq x_{p+1,j} \quad \forall p \in \{1, \ldots, nd-1\}, \forall i, j \in M \tag{5}$$

$$x_{p,i} + x_{p+1,j} - 1 \leq y_{p,i,j} \quad \forall p \in \{1, \ldots, nd-1\}, \forall i, j \in M \tag{6}$$

$$y_{nd,i,j} \leq x_{nd,i} \quad \forall i, j \in M \tag{7}$$

$$y_{nd,i,j} \leq x_{1,j} \quad \forall i, j \in M \tag{8}$$

$$x_{nd,i} + x_{1,j} - 1 \leq y_{nd,i,j} \quad \forall i, j \in M \tag{9}$$

$$x_{p,i} \in \{0, 1\}, y_{p,i,j} \in \{0, 1\} \tag{10}$$

In assembly lines, a steady supply of parts to stations is important to prevent bottlenecks, avoid shortages, and reduce in-process inventories. The second mode (Model 2) takes the supply of parts to stations into account by optimizing workload equalization, which is achieved by minimizing violations of part-frequency constraints. Here, part-frequency constraints denote that, for one sliding window of $B_l$ consecutive product positions, there are at most $A_l$ products requiring part $l$ within the $B_l$ consecutive positions. The violations of part-frequency constraints could disrupt material flow, increase logistics complexity, and result in uneven station loads. Hence, the minimization of violations of part-frequency constraints is important in real applications to promote smoother production and better resource utilization. The decision variables utilized in Model 2 are defined below.

| Variable | Domain | Description |
|---|---|---|
| $x_{p,i}$ | $\{0, 1\}$ | 1, if the product at position $p$ is of type $i$; 0, otherwise. |
| $q_{p,l}$ | $\{0, 1\}$ | 1, if part-frequency constraint for part $l$ is violated at position $p$; 0, otherwise. |
| $u_{p,l,b}$ | $\{0, 1\}$ | 1, if part $l$ is used at the $b$-th position in the window starting at position $p$; 0, otherwise. |

Model 2 minimizes violations of part-frequency constraints. Equation (11) sums all violations. Constraints (12)–(13) are the same assignment and demand constraints as in Model 1. Equation (14) computes part usage within sliding windows using modulo arithmetic. Constraints (15)–(16) use the big-M method to detect violations: if usage exceeds $A_l$, then $q_{p,l} = 1$. The linearization is detailed in Section 3.3.

$$\text{Min } F_2 = \sum_{p=1}^{nd} \sum_{l \in R} q_{p,l} \tag{11}$$

$$\sum_{i \in M} x_{p,i} = 1 \quad \forall p \in D \tag{12}$$

$$\sum_{p \in D} x_{p,i} = d_i \quad \forall i \in M \tag{13}$$

$$u_{p,l,b} = \sum_{i \in M} z_{i,l} \cdot x_{(p+b-1) \bmod nd, i} \quad \forall p \in D, \forall l \in R, \forall b \in \{1, \ldots, B_l\} \tag{14}$$

$$\sum_{b=1}^{B_l} u_{p,l,b} - A_l \leq BigM \cdot q_{p,l} \quad \forall p \in D, \forall l \in R$$

(15)

$$A_l - \sum_{b=1}^{B_l} u_{p,l,b} \leq BigM \cdot (1 - q_{p,l}) \quad \forall p \in D, \forall l \in R$$

(16)

$$x_{p,i} \in \{0, 1\}, q_{p,l} \in \{0, 1\}, u_{p,l,b} \in \{0, 1\}$$

(17)

In mixed-model and reconfigurable lines, it is often desirable to have a constant or nearly constant rate of output for each product variant (a concept known as "rate smoothing" or "level scheduling"). This constant rate aims at stabilizing downstream processes, including part supply, material handling, and delivery schedules. The third mode (Model 3) considers this situation to optimize logistics leveling, which is achieved by minimizing the deviation of actual cumulative production from an ideal and evenly-spaced production rate for each product type. Model 3 quantifies and minimizes these deviations, and the decision variables for Model 3 are introduced as follows.

| Variable | Domain | Description |
|---|---|---|
| $x_{p,i}$ | $\{0, 1\}$ | 1 if the product at position $p$ is of type $i$; 0 otherwise. |
| $t_{p,k}$ | $\mathbb{R}^+$ | Completion time of product at position $p$ at station $k$. |
| $u_{p,i}$ | $\mathbb{R}^+$ | Cumulative number of product $i$ up to position $p$. |
| $v_{p,i}$ | $\mathbb{R}^+$ | Ideal cumulative number of product $i$ at position $p$. |
| $\delta_{p,i}^+, \delta_{p,i}^-$ | $\mathbb{R}^+$ | Positive and negative deviations from the ideal cumulative count. |
| $T$ | $\mathbb{R}^+$ | Total completion time (makespan). |

Model 3 minimizes logistics leveling by reducing deviation from ideal production rates. Equation (18) linearizes the absolute deviation. Constraints (19)–(20) are the assignment and demand constraints. Equations (21)–(24) compute station completion times with inter-station intervals. Constraints (25)–(26) calculate cumulative counts. Equation (27) defines makespan $T$. Constraint (28) contains a bilinear term $v_{p,i} \cdot T = d_i \cdot t_{p,nk}$, which is handled by a two-stage ε-constraint (see Section 3.3). Equation (29) linearizes the deviation using positive and negative slack variables. Notice that constraint (28), $v_{p,i} \cdot T = d_i \cdot t_{p,nk}$, contains a bilinear term ($v_{p,i} \times T$), rendering the model a bilinear program. To ensure exact solvability via standard MILP solvers and to establish a reliable benchmark, this study implements an ε-constraint method in two sequential linear stages. Stage 1 minimizes the total completion time $T$ subject to constraints (19)–(27) and the original assignment constraints, obtaining its optimal value $T^*$. Afterwards, fixing $T = T^*$ in constraint (28) (which then becomes linear), Stage 2 solves the resulting pure MILP to minimize the total absolute deviation (objective [18]). This decomposition guarantees that both sub-models are linear and can be solved to optimality. The two-stage procedure ensures that the obtained solution is Pareto-optimal with respect to the third objective, while maintaining linearity in each stage.

$$\text{Min } F_3 = \sum_{p=1}^{nd} \sum_{i \in M} \left( \delta_{p,i}^+ + \delta_{p,i}^- \right)$$

(18)

$$\sum_{i \in M} x_{p,i} = 1 \quad \forall p \in D$$

(19)

$$\sum_{p \in D} x_{p,i} = d_i \quad \forall i \in M$$

(20)

$$t_{1,1} = \sum_{i \in M} x_{1,i} \cdot PT_{i,1} \tag{21}$$

$$t_{p,1} = t_{p-1,1} + IT + \sum_{i \in M} x_{p,i} \cdot PT_{i,1} \quad \forall p \in \{2, \ldots, nd\} \tag{22}$$

$$t_{p,k} \geq t_{p,k-1} + \sum_{i \in M} x_{p,i} \cdot PT_{i,k} \quad \forall p \in D, \forall k \in \{2, \ldots, nk\} \tag{23}$$

$$t_{p,k} \geq t_{p-1,k} + \sum_{i \in M} x_{p,i} \cdot PT_{i,k} \quad \forall p \in \{2, \ldots, nd\}, \forall k \in \{2, \ldots, nk\} \tag{24}$$

$$u_{1,i} = x_{1,i} \quad \forall i \in M \tag{25}$$

$$u_{p,i} = u_{p-1,i} + x_{p,i} \quad \forall p \in \{2, \ldots, nd\}, \forall i \in M \tag{26}$$

$$T = t_{nd,nk} \tag{27}$$

$$v_{p,i} \cdot T = d_i \cdot t_{p,nk} \quad \forall p \in D, \forall i \in M \tag{28}$$

$$u_{p,i} - v_{p,i} = \delta_{p,i}^+ - \delta_{p,i}^- \quad \forall p \in D, \forall i \in M \tag{29}$$

$$x_{p,i} \in \{0, 1\}, t_{p,k} \geq 0, u_{p,i} \geq 0, v_{p,i} \geq 0, \delta_{p,i}^+ \geq 0, \delta_{p,i}^- \geq 0, T \geq 0 \tag{30}$$

### 3.3 Analysis of the formulated models

The three MILP sub-models presented in Section 3.2 are designed to be directly solvable by standard commercial solvers, thereby providing exact benchmarks for small-to-medium instances. To highlight the advancements achieved, this section compares the proposed formulations with prior work, particularly the models introduced by Yuan, Deng [13].

The formulations in Yuan, Deng [13] serve as a conceptual framework for the RALSP but suffer from fundamental limitations that prevent direct solution by MILP solvers. 1) Their workload equalization model defines the objective via a conditional rule $p_{m,n} = 1$ if $\left[H_n - \sum_i \sum_t a_{i,m} x_{n+t,i} \geq 0\right]$, which cannot be expressed as linear constraints without auxiliary binary variables and big-M transformations. 2) Their delayed workload model involves max() functions in constraints (8) and (9), which are inherently nonlinear and non-convex and cannot be handled by MILP solvers without omitted linearization steps. 3) Moreover, the three sub-models use inconsistent variable definitions and constraint structures, making it impossible to combine them into a single multi-objective MILP model that can be solved simultaneously. In contrast, the models in this study share a common variable set and can be jointly addressed through $\epsilon$-constraint or weighted-sum methods.

Consequently, while the earlier work provides valuable insights into the problem structure, it does not yield a solvable MILP formulation. The models proposed in this study overcome these limitations through complete linearization strategies: Model 1 linearizes product-of-binary terms using auxiliary variables $y_{p,i,j}$ and linear constraints (4)–(9); Model 2 encodes part-frequency constraint violations using the big-M method, transforming the original logical condition into linear inequalities (15)–(16); and Model 3 handles the bilinear term in constraint (28) via a two-stage $\epsilon$-constraint approach, while linearizing absolute deviations using positive and negative slack variables. These linearizations ensure that all three models are directly solvable by commercial solvers such as CPLEX and Gurobi, providing exact optimal solutions for small-to-medium instances and reliable benchmarks for evaluating heuristic algorithms. For practical multi-objective optimization, the linear models can be combined using weighted-sum or $\epsilon$-constraint methods, maintaining computational tractability.

## 4. Proposed methodology

To address the multi-objective RALSP formulated in Section 3, this study proposes a novel QMOHH. The algorithm employs a Q-learning controller to select from a pool of eight heuristic operators adaptively. The heuristic operators are derived from four established metaheuristics, including PSO, TLBO, WOA, and GWO. Meanwhile, to enhance search performance, QMOHH employs a new density-aware leader selection strategy with a survival-time decay factor to select the global best solution for population evolution. The new density-aware leader selection strategy ensures effective utilization of high-quality solutions while preventing over-reliance on any single individual.

### 4.1 Main procedure of QMOHH

The framework of the proposed QMOHH is outlined in Algorithm 1. QMOHH begins by initializing a population, along with a Pareto archive for maintaining non-dominated solutions. A Q-learning agent is also initialized to guide operator selection.

During each iteration, the algorithm determines its current state based on evaluation progress and uses an ε-greedy policy to select one of the eight heuristic operators. If required, a global best solution is chosen from the archive using the new density-aware leader selection strategy. The selected operator is employed to generate new solutions, and subsequently, the new solutions are evaluated. Afterwards, the personal best solutions and Pareto archive are updated when necessary. A reward—based on the number of newly added non-dominated solutions—is computed and used to update the Q-table. This main loop is repeated until the termination criterion (a maximum number of evaluations in this study). After completing the main loop, the Pareto archive is returned as the achieved approximate Pareto-optimal set by QMOHH. With the Q-learning and new density-aware leader selection strategy, QMOHH achieves the proper balance between exploration and exploitation throughout the search process.

### Algorithm 1. Proposed Q-learning-based multi-objective hyper-heuristic

```
Input: Instance data, parameters (population size N and maximum evaluations)
Output: Achieved an approximate Pareto archive
1: Initialize population P randomly under the demand constraints;
2: Evaluate the initial population, where the objective values for each solution are calculated;
3: Update Pareto archive with the initial population;
4: Update personal best solutions (P_best) with P_best←P;
5: Create states, actions, and a Q-table to initialize the Q-learning agent;
6: Achieve current state based on the current progress (evaluations/max_evaluations);
7: While evaluations<max_evaluations do
8:   Achieve the current state based on evaluations/max_evaluations;
9:   Achieve the selected operator for population evolution with Q-learning;
10:   If the selected operator requires a global leader then
11:     Achieve global best with the density-aware leader selection strategy from the Pareto archive;
12:   End-if
```

```
13:    Obtain a new population (P_new) with the selected operator;
14:    Evaluate the new population and update the evaluation count;
15:    Update personal best solutions (P_best) if the new solution dominates the current solution;
16:    Update Pareto archive with non-dominated solutions from P_new;
17:    reward ←the number of newly added non-dominated solutions;
18:    next_state←determine next state based on the evaluation progress;
19:    update Q_table;
20:    P←select the N solutions for P and P_new;
21:    current_state ← next_state;
22: End-while
23: Return Pareto archive// Final Pareto-optimal solution set
```

### 4.2 Solution presentation

A product sequence of length $nd = \sum_i d_i$ is necessary for RALSP to achieve objective values. In product sequence, each position corresponds to a product. Here, demand constraints $d_i$ for each product type $i$ must be satisfied in this product sequence. Suppose that there are four product types (A, B, C, D) with demands [1–3]. One feasible sequence is [A, B, C, A, C, D, B, C].

The proposed QMOHH utilizes a string of floating-point numbers with the size of $nd$ for encoding. Based on the floating-point numbers, the random-key method is applied to achieve a feasible product sequence. Fig 3 provides an example solution presentation, where there are four product types (A, B, C, D) with a minimum production cycle of [1–3]. Eight floating-point numbers are randomly generated within the range [0.00, 1.00] for encoding, and the original task sequence is [A, A, B, B, C, C, C, D]. After sorting the floating-point numbers in ascending order, the final product sequence could be achieved, namely [B, C, C, B, A, A, D, C].

Once the product sequence is determined by sorting the floating-point numbers in encoding, three objectives could be calculated directly on the basis of the three linearized MILP models in Section 3.2, respectively. Specifically, reconfiguration cost is achieved by summing the cost incurred between consecutive product pairs in the task sequence. Production workload equalization is calculated by summing the violations of the part-frequency constraints along the product sequence. Logistics leveling is calculated by summing the deviation of the actual cumulative production from the ideal, evenly-spaced production rate for each product type. Clearly, the demand constraints $d_i$ for each product type $i$ are satisfied for any string of floating-point numbers with the size of $nd$. This floating-point encoding scheme ensures the generation of feasible sequences throughout the search process and facilitates the application of the various metaheuristic operators described in Section 4.3.

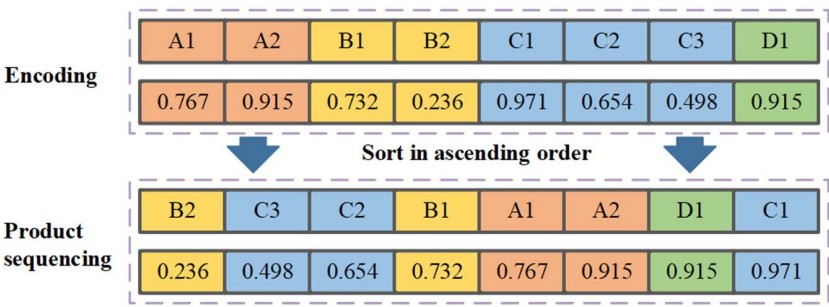

**Fig 3. Solution presentation.**

## 4.3 Metaheuristic operators

The proposed QMOHH employs eight heuristic operators derived from four well-established metaheuristics. Each heuristic operator is adapted to modify the floating-point numbers in encoding, aiming at obtaining diverse and feasible product sequences.

PSO-based operators: Three operators are adapted from PSO [36]. These include the first operator with basic position update, the second operator combining position update and uniform mutation to emphasize exploration, and the third operator combining position update and non-uniform mutation. The third operator utilizes the non-uniform mutation with time-decreasing strength to achieve phased refinement.

TLBO-based operators: Three operators are derived from TLBO [37]. These include the teacher-phase operator simulating a teacher guiding the students, the learning-phase operator simulating peer-to-peer learning among students, and the mixed TLBO phase combining the teacher phase and learning phase to integrate global and local search.

WOA-based operator: This operator implements the three characteristic behaviors of WOA—encircling prey, bubble-net attacking, and random search—probabilistically [38]. Adaptive parameters shift the emphasis from exploration to exploitation throughout the optimization process.

GWO-based operator: Based on the GWO [39], this operator employs a hierarchy of three leading solutions (alpha, beta, delta) selected from the archive to guide population updates. The implementation of three leaders emphasizes the search around the space between Pareto solutions.

Notice that the global best solution is necessary for all the heuristic operators except for the learning-phase operator in multi-objective optimization, and the new density-aware leader selection strategy could be utilized to obtain the global best solution in iterations. All operators are utilized to modify the continuous encoding scheme. And any new continuous encoding scheme could be transferred into a feasible product sequence utilizing the decoding in Section 4.2, and hence, the demand constraints are satisfied.

Among the eight heuristic operators, some operators emphasize global exploration while others emphasize local refinement. They provide the Q-learning mechanism with a rich set of strategies to dynamically select from, and hence enable effective adaptation to the evolving solution space.

## 4.4 Q-Learning mechanism for operator selection

Q-learning is a model-free, value-based reinforcement learning algorithm that has been widely utilized in the literature. Q-learning operates by estimating the expected cumulative reward (Q-value) for taking a specific action in a given state [31]. The algorithm has a Q-table where each entry $Q(s, a)$ contains state $s$ and action $a$, indicating the expected utility of choosing action $a$ in state $s$. The Q-values are updated using the expression [31] in iterations, where $\alpha$ is the learning rate, $\gamma$ is the discount factor, $r_t$ is the immediate reward received after taking action $a_t$ in state $s_t$, and $s_{t+1}$ is the next state.

$$Q_{t+1}(s_t, a_t) \leftarrow Q_t(s_t, a_t) + \alpha \left[ r_t + \gamma \max_a Q_t(s_{t+1}, a) - Q_t(s_t, a_t) \right]$$

(31)

In the proposed QMOHH algorithm, Q-learning is responsible for selecting one of eight heuristic operators at each iteration adaptively. Q-learning starts with determining state representation, action set, and reward function, and in the main loop, Q-learning conducts the action selection and policy update iteratively.

**State representation:** The state is defined based on the progress of the optimization process. The algorithm's evaluation progress, calculated as the ratio of completed evaluations (*evaluations*) to the maximum allowed evaluations (*max_evaluations*), is discretized into ten intervals (0.0 to 0.9). This discretization creates a state space that effectively captures the different stages of the search, allowing the Q-learning policy to be phase-dependent. Unlike approaches that rely on problem-specific metrics (e.g., population diversity, temperature), this state definition is problem-independent and requires no instance-specific tuning, facilitating generalization to other scheduling problems.

**Action set:** The action set contains eight elements corresponding to eight heuristic operators in Section 4.3, including three PSO-based operators, three TLBO-based operators, one WOA-based operator, and one GWO-based operator.

**Reward function:** After applying the selected operator and evaluating the new populations, the reward is calculated as the number of new non-dominated solutions added to the external Pareto archive from the newly generated population. This design directly aligns the Q-learning objective with the goal of multi-objective optimization—expanding the Pareto front—rather than relying on scalar fitness improvements commonly used in single-objective or parameter-tuning contexts.

**Action selection and policy update:** At each iteration, the current state is determined, and later one heuristic operator is selected based on the Q-table utilizing ε-greedy policy. After generating the new population with the selected heuristic operator, the new population is evaluated, and the reward is computed. Later on, the Q-table is updated using the standard Q-learning update rule based on the reward.

### 4.5 New density-aware leader selection strategy

A global best solution (*gbest*) is necessary for population evolution in iterations, and the method of selecting *gbest* from the Pareto archive is crucial for guiding the search direction. Different from the methods in Coello, Pulido [36], the proposed QMOHH employs a new density-aware leader selection strategy that incorporates a survival-time decay factor. The strategy calculates a modified crowding distance $\widehat{CD_i}$ for each archive solution as $\widehat{CD_i} = CD_i \times \gamma^{\tau_i}$, where $CD_i$ is the original crowding distance in Deb, Pratap [40], $\tau_i$ is the solution's selection time in iterations, and $\gamma$ is a decay rate (e.g., 0.5). This formula reduces the attractiveness of long-surviving solutions, even if they reside in sparse regions. The survival-time decay factor $\gamma^{\tau_i}$ can be intuitively understood as a "freshness" penalty: a solution that has been selected as the global leader many times becomes progressively less likely to be chosen again, giving newer or less frequently selected solutions an opportunity to guide the population. This mechanism prevents the algorithm from getting trapped around a few dominant individuals and promotes continuous exploration of sparser regions of the Pareto front. Hence, this new density-aware leader selection strategy encourages exploration of new areas and prevents search stagnation. The leader selection process is illustrated in Fig 4.

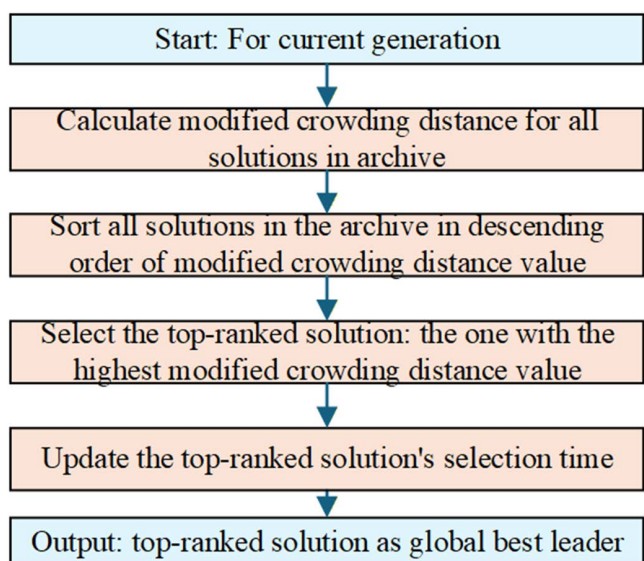

**Fig 4. Flowchart of the enhanced global best (*gbest*) selection process.**

The survival-time decay factor reduces the selection probability of solutions that have been repeatedly chosen as leaders, ensuring a balanced exploration of the Pareto front. This mechanism ensures diversity-driven exploration by consistently favoring leaders from less crowded regions of the Pareto front, while the decay factor automatically balances this with the need to exploit known high-quality areas over time.

## 5. Numerical case study

In this section, the developed models and QMOHH are utilized to solve a real-world case introduced by Yuan, Deng [13]. This case is taken from Changzhou AMEC&GBM Motor Company, which procures a family of DC electric motors in a reconfigurable assembly line. Similar to other reconfigurable systems, the assembly line in this case has the following challenges: producing multiple product variants in small batches while pursuing the minimization of changeover effort and the maintenance of smooth production flow.

This real-world case consists of five stations ($nk = 5$) to produce 4 distinct motor models belonging to the same product family. The motor models are: 62ZYT001, 100ZYT001, 62ZYT-SUV, and 78ZYT001, denoted as products A, B, C, and D, respectively. The demand for products is [3–5] under the minimum production cycle, which results in a product sequencing problem with $d = 5 + 4 + 4 + 3 = 16$ products. Relevant production data are summarized in Table 1 (reconfiguration cost matrix, in thousand CNY), Table 2 (part-requirement matrix, where "1" indicates the part is required), and Table 3

**Table 1. Reconfiguration cost matrix between product types.**

| Product | Reconfiguration cost (×1000 CNY) | | | |
|---|---|---|---|---|
| | To A | To B | To C | To D |
| A | 0 | 2 | 4 | 4 |
| B | 3 | 0 | 5 | 6 |
| C | 5 | 3 | 0 | 4 |
| D | 3 | 2 | 7 | 0 |

**Table 2. Part-requirement matrix for each product type.**

| Product | Part requirement | | | |
|---|---|---|---|---|
| | Part P1 | Part P2 | Part P3 | Part P4 |
| A | 1 | 1 | 0 | 0 |
| B | 0 | 1 | 0 | 1 |
| C | 1 | 0 | 1 | 0 |
| D | 0 | 1 | 1 | 0 |

**Table 3. Operation time of each product type at each station (unit: time-units).**

| Product | Operation times at stations | | | | |
|---|---|---|---|---|---|
| | Station 1 | Station 2 | Station 3 | Station 4 | Station 5 |
| A | 5 | 8 | 4 | 6 | 8 |
| B | 7 | 7 | 9 | 5 | 6 |
| C | 8 | 5 | 5 | 9 | 4 |
| D | 6 | 9 | 7 | 4 | 7 |

(operation time per station, in time-units). The following part-frequency constraints apply, expressed as $(A_l; B_l)$: Part P1: (2; 3); Part P2: (3; 5); Part P3: (3; 4); Part P4: (1; 2). A fixed interval of 3 time-units is assumed between successive product startups.

To solve this multi-objective problem with the model, the ε-constraint method is first utilized to obtain Pareto-optimal solutions. Firstly, each of the three MILP models (Model 1, Model 2, Model 3) is solved individually to obtain its minimum value. Based on the achieved product sequences by the models, the values of all objectives are achieved and the maximum value for each objective is approximated by using the worst values. Secondly, a grid of ε values is created for two of the objectives while the third is taken as the primary objective. To generate the Pareto set, an interval step of 5 is utilized: the range of each objective is divided into 5 equal parts, yielding 6 ε values per objective. Including a relaxed case (ε = ∞) for each constraint, a total of 7 × 7 = 49 subproblems are solved for a given primary objective, and cycling through all three primary objectives leads to 3 × 49 = 147 MILP runs. Here, the step size can be adjusted to obtain a finer or coarser approximation of the Pareto frontier. Thirdly, a set of MILP models are solved under many combinations of objective values. If the first objective is the primary objective, the MILP is solved: min $f_1$ s.t. $f_2 \leq \varepsilon_2$, $f_3 \leq \varepsilon_3$, and the linear constraints of Models for each combination $(\varepsilon_2, \varepsilon_3)$. Here, the bilinear term $v_{p,i} \cdot T = d_i \cdot t_{p,nk}$ is handled by a two-stage linearization in Model 3. Finally, all obtained solutions are merged, and a dominance filter is applied to extract the non-dominated set.

All the models terminate when the optimal solution is found or the computation time reaches 3600 seconds (s). Table 4 presents the 13 Pareto-optimal solutions obtained via the ε-constraint method. As seen in this table, reconfiguration cost ranges from 13 to 67, workload violation ranges from 7 to 24, and workload equalization ranges from 22.51 to 64.23. The solution with the lower reconfiguration cost (e.g., 13) has the higher workload violation and higher logistics leveling deviation; the solution with reduced workload violation or logistics leveling required substantially higher reconfiguration cost. These findings confirm the evident trade-offs among the objectives. This set of diverse non-dominated solutions provides many scheduling alternatives for the decision-maker to select, demonstrating the effectiveness of the formulated models.

The proposed QMOHH algorithm is subsequently applied and the best solution in 10 runs are recorded, where 85 non-dominated solutions are yielded. As seen in Fig 5, QMOHH could expand the discovered Pareto frontier in the search regions. This demonstrates QMOHH's strength in broad exploration and its ability to approximate an extensive and well-distributed frontier. Notice that some of the Pareto solutions by the developed model are dominated by the Pareto solutions by QMOHH. The reason lies behind two factors: firstly, the discrete interval step used in the ε-constraint method limits the

**Table 4. Pareto solutions achieved by the developed model.**

| Index | Product sequence | $F_1$ | $F_2$ | $F_3$ |
|---|---|---|---|---|
| 1 | [A, A, A, A, C, C, C, C, D, D, D, B, B, B, B, A] | 13 | 24 | 64.22857 |
| 2 | [C, C, C, D, D, D, B, B, A, A, A, A, A, B, B, C] | 16 | 21 | 59.69591 |
| 3 | [C, C, C, B, B, A, A, A, A, A, D, D, D, B, B, C] | 17 | 18 | 53.1462 |
| 4 | [C, C, B, B, A, A, A, D, D, D, B, B, A, A, C, C] | 19 | 18 | 39.96491 |
| 5 | [D, D, A, A, B, B, C, C, C, C, B, B, A, A, A, D] | 20 | 16 | 43.65714 |
| 6 | [A, A, D, D, B, A, A, B, C, C, D, A, B, C, C, B] | 36 | 7 | 47.04 |
| 7 | [C, D, A, A, B, D, D, A, A, B, C, C, B, A, B, C] | 38 | 7 | 40.35088 |
| 8 | [C, D, B, A, A, B, A, C, C, D, B, A, D, B, A, C] | 40 | 10 | 25.80117 |
| 9 | [D, A, B, C, C, B, D, A, A, B, D, A, A, B, C, C] | 44 | 7 | 31.87135 |
| 10 | [C, D, A, B, C, C, B, A, D, A, B, A, D, A, B, C] | 46 | 7 | 31.62573 |
| 11 | [B, A, D, B, A, A, D, C, C, B, A, D, C, C, B, A] | 48 | 7 | 29.66857 |
| 12 | [A, B, C, D, A, B, C, A, D, B, C, A, D, B, A, C] | 60 | 13 | 22.60819 |
| 13 | [A, B, D, C, A, B, C, A, D, C, B, A, D, B, A, C] | 67 | 12 | 22.51462 |

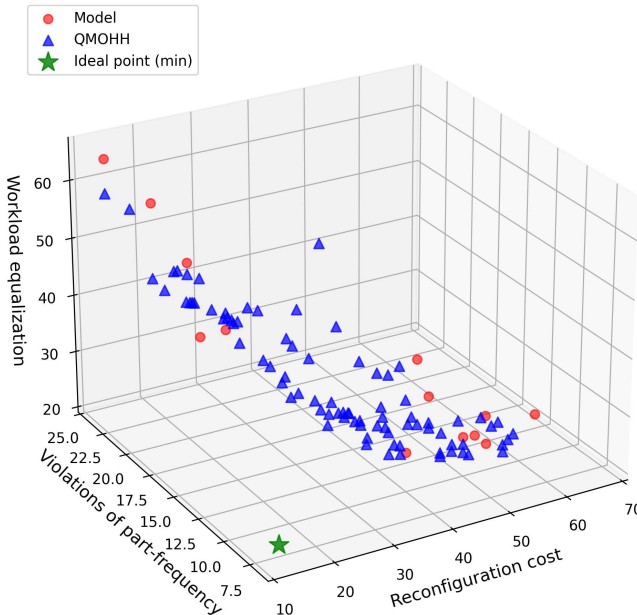

**Fig 5. Pareto solutions obtained by the proposed QMOHH and the developed model.**

exploration of the continuous trade-off space; secondly, solving the multi-objective problem requires a two-stage decomposition due to the bilinear term in Model 3, which may further restrict the search space for the third objective in the final stage.

Meanwhile, the average computation time in each run is 1.78s and the total computation time is 261.99s. For the QMOHH, the computation time is 8.06s in each run on average and the total computation time is 80.59s. In short, this case study first verifies that the developed mathematical model can generate a set of high-quality Pareto solutions, which capture the fundamental trade-offs among the objectives. Meanwhile, QMOHH has a strong exploration capability in navigating the complex multi-objective landscape, and it uncovers a broad range of high-quality trade-off solutions with less computation time. This case study demonstrates that QMOHH is capable of providing decision-makers with a comprehensive approximation of the true Pareto-optimal set.

## 6. Experimental results

This section conducts a comparative study to evaluate the performance of the proposed QMOHH. Firstly, Section 6.1 introduces the solved instances and the compared algorithms. Next, Section 6.2 evaluates the model with ε-constraint method and Section 6.3 evaluates the performance improvements of the proposed QMOHH. Finally, Section 6.4 presents the comparative study, where the proposed QMOHH is compared with nine other multi-objective algorithms.

### 6.1 Experimental design

To evaluate the proposed QMOHH, 120 test instances are generated based on realistic production scenarios. Each instance consists of four product types. The minimum production cycle (product demands) is defined using 20 pre-determined patterns, with total demand per cycle ranging from 10 to 29. Each pattern satisfies that the greatest common divisor of the four demands is 1, ensuring that no further reduction is needed. The number of stations takes six fixed values: 5, 10, 15, 20, 25, and 30, each producing 20 instances. The processing time for each product at a given station

is uniformly generated between 6 and 25 time-units. Reconfiguration costs are generated as an $nm \times nm$ matrix with zero diagonal and off-diagonal entries uniformly drawn from 0 to 12 (in thousand CNY). Part requirements are generated by first creating a random integer between 0 and 8 for each product-part pair; the binary requirement $z_{i,l}$ is set to 1 if this integer > 0, otherwise 0. The number of part types per instance is randomly set between 12 and 16. For part-frequency constraints, the window length $B_l$ is randomly selected from $\{3, 4, 5\}$ and the maximum allowable count $A_l$ is randomly chosen between 1 and $B_l$. The interval time between successive product startups is fixed to 5 time-units. Instances are classified by scale according to the number of stations: instances with 5 or 10 stations are categorized as small-scale; those with 15 or 20 stations as medium-scale; and those with 25 or 30 stations as large-scale. This instance set comprises 40 small-scale, 40 medium-scale, and 40 large-scale instances, for a total of 120 instances. All instances, along with the source code of all algorithms, are publicly available at https://github.com/zixiangliwust/Instances_RALSP under the MIT license.

Three indicators are used to evaluate the Pareto solution sets: GD (generational distance) measures convergence to the reference front; $I_\epsilon$ (ε-indicator) quantifies the smallest factor needed for the obtained set to dominate the reference; and 1-NHV (complement of normalized hypervolume) assesses both convergence and diversity. Smaller values indicate better performance for all indicators. These indicators have been widely applied in the literature [41,42] and detailed calculation procedures are available in the source code on GitHub (see Data Availability).

To evaluate the performance of the proposed QMOHH, it is compared with nine other multi-objective algorithms, including multi-objective simulated annealing algorithm (MOSA) [43], multi-objective restarted simulated annealing (MRSA) [41], multi-objective artificial bee colony algorithm (MOABC) [44], improved multi-objective artificial bee colony (IMOABC) [42], GDE3 [45], MOEAD [46], NSGA-II [40], multi-objective PSO algorithm (MOPSO) [36], a new PSO-based metaheuristic (SMPSO) [47].

The termination criterion for all tested algorithms is a total of 100,000 evaluations. The proposed linearized MILP models are solved using the IBM ILOG CPLEX Optimization Studio (version 22.1.1) solver. And all tested algorithms are implemented in Python 3.10 on a personal computer equipped with an Intel(R) Core (TM) Ultra 9 185H, with parallel computation employed to accelerate the experiments. For each independent run, the random seed is set to the current system time to ensure statistical independence across the 10 runs per instance.

All algorithms are calibrated using the Taguchi experimental design method to ensure fair comparison. A set of 30 representative instances (covering small, medium, and large scales) is used for calibration. For each algorithm, an $L_{16}(4^5)$ orthogonal array is employed to evaluate five key parameters at four levels. The response metric is the average 1-NHV over 10 independent runs. The optimal parameter combination for each algorithm is selected based on the signal-to-noise ratio. The final calibrated parameters for QMOHH are: population size $N = 100$, learning rate $\alpha = 0.1$, discount factor $\gamma = 0.95$, initial ε-greedy exploration probability $\varepsilon = 0.9$ (decaying linearly to 0.1), survival-time decay factor $\gamma_{decay} = 0.5$, uniform and non-uniform mutation probabilities $1.0/nd$ with perturbation 0.5 and termination at $100,000$ evaluations. These values are explicitly defined in the source code, which is publicly available at https://github.com/zixiangliwust/Instances_RALSP. The same calibration procedure is applied to the nine compared algorithms; their final parameter settings can also be found in the code.

### 6.2 Evaluating the MILP model with ε-constraint method

The results in Table 5 illustrate the computational behavior of the MILP models under the ε-constraint method with a time limit of 300s per subproblem. Here, total time(s) reports the total computation time. From this table, it is observed that, as the number of stations ($nk$) or the total number of products ($nd$) increases, the total runtime rises substantially. For small instances (e.g., $nk = 5$, $nd = 10$), all 147 ε-combinations are solved to optimality within 20s, yielding 40 valid solutions and 3 Pareto points. In contrast, for the largest instance ($nk = 30$, $nd = 29$), the total runtime exceeds 7888s, and only 5 out of 147 subproblems reach optimality. Instances with larger $nd$ (i.e., longer product sequences) consistently require longer

**Table 5. Computational results of the ε-constraint method on selected instances.**

| Instance | nk | nd | Total combinations | #Valid solutions | #Optimal solutions | #Pareto solutions | Total Time(s) |
|---|---|---|---|---|---|---|---|
| instance_001 | 5 | 10 | 147 | 40 | 40 | 3 | 19.71 |
| instance_011 | 5 | 19 | 147 | 37 | 37 | 16 | 201.17 |
| instance_020 | 5 | 29 | 147 | 24 | 13 | 19 | 3744.45 |
| instance_021 | 10 | 10 | 147 | 36 | 36 | 4 | 23.39 |
| instance_031 | 10 | 19 | 147 | 16 | 16 | 6 | 82.35 |
| instance_040 | 10 | 29 | 147 | 10 | 10 | 6 | 592.88 |
| instance_041 | 20 | 10 | 147 | 48 | 48 | 4 | 29.63 |
| instance_051 | 15 | 19 | 147 | 21 | 21 | 7 | 124.01 |
| instance_060 | 15 | 29 | 147 | 8 | 3 | 7 | 2856.47 |
| instance_061 | 15 | 10 | 147 | 49 | 49 | 5 | 37.75 |
| instance_071 | 20 | 19 | 147 | 10 | 10 | 5 | 448.6 |
| instance_080 | 20 | 29 | 147 | 10 | 4 | 9 | 2640.03 |
| instance_081 | 20 | 10 | 147 | 46 | 46 | 2 | 35.03 |
| instance_091 | 25 | 19 | 147 | 9 | 9 | 4 | 953.75 |
| instance_100 | 25 | 29 | 147 | 12 | 6 | 4 | 2023.58 |
| instance_101 | 30 | 10 | 147 | 52 | 52 | 7 | 55.77 |
| instance_111 | 30 | 19 | 147 | 10 | 10 | 6 | 1557.36 |
| instance_120 | 30 | 29 | 147 | 8 | 5 | 5 | 7888.91 |

solution times, even when the number of stations remains moderate. For instance, with $nk = 10$, increasing $nd$ from 10 to 29 raises the total runtime from 23s to nearly 600s, while the number of optimal subproblems drops from 36 to 10.

The computational cost becomes prohibitive when both $nk$ and $nd$ exceed moderate values. Consequently, the MILP models are best suited for small-scale instances (e.g., $nk \leq 10$, $nd \leq 20$) where exact Pareto frontiers can be obtained within acceptable time. For larger problems, the use of metaheuristics such as QMOHH is essential. Moreover, finer ε-grids (e.g., smaller interval steps) would further increase the number of subproblems, making the approach even more computationally demanding.

These findings confirm that the developed MILP formulations provide exact benchmarks for small-to-medium instances, while larger-scale problems require efficient meta-heuristic methods.

## 6.3 Evaluating the improvements of QMOHH

To validate the contribution of each proposed component, the complete QMOHH is compared against four variants, as defined below. All four variant algorithms, along with the complete QMOHH, are evaluated on the 120 benchmark instances described in Section 6.1, following the same experimental protocol.

**MOHH-v1:** Only three PSO-based operators are utilized. It is utilized to establish a PSO-centric baseline to assess the value of the utilization of eight heuristic operators.

**MOHH-v2:** Q-learning mechanism is disabled, and heuristic operators are randomly selected. It is utilized to assess the value of adaptive operator selection with the Q-learning mechanism.

**MOHH-v3:** Q-learning controller is replaced with a Deep Q-Network (DQN). It is utilized to examine the impact of a different reinforcement learning architecture.

**MOHH-v4:** The standard method to select the global best solution from MOPSO is utilized to replace the new density-aware leader selection strategy. It is utilized to test the advantage of the new density-aware leader selection strategy.

In this section, the 1-NHV indicator is utilized to measure the performance of these five algorithms, where the smaller value of 1-NHV denotes the better overall solution quality. The average 1-NHV values over the 120 instances are presented in Table 6. The Friedman test is conducted on the mean ranks of the algorithms for each performance indicator. Fig 6 depicts the means plot of the average ranks of MOHH methods in terms of the 1-NHV indicator.

The results in Table 6 verify the clear superiority of QMOHH, which achieves the lowest average 1-NHV of 0.232. MOHH-v1 achieves worse performance (0.290) compared to QMOHH, underscoring the benefit of integrating a diverse set of metaheuristic operators. Again, MOHH-v2 achieves worse performance (0.248) compared to QMOHH, indicating that disabling the Q-learning mechanism leads to a noticeable performance degradation. This finding highlights the critical role of adaptive operator selection with the Q-learning mechanism. Notably, MOHH-v3 exhibits the poorest performance among the compared methods, suggesting that the Q-learning framework is more suitable and efficient than a more complex deep reinforcement learning approach for online heuristic operator selection. MOHH-v4 obtains worse performance (0.242) compared to QMOHH, confirming the effectiveness of the new density-aware leader selection strategy.

This ablation study conclusively validates that each core component of QMOHH—the Q-learning-based adaptive operator selection, the ensemble of diverse heuristics, and the new density-aware leader selection strategy—contributes uniquely and essentially to its superior search capability and final solution quality for the multi-objective RALSP.

## 6.4 Comparative study

To comprehensively evaluate the performance of the proposed algorithm, QMOHH is compared with nine other state-of-the-art multi-objective algorithms in Section 6.1. All algorithms are evaluated using the same 120 benchmark instances described in Section 6.1, under the identical termination criterion. Each algorithm is executed 10 times on each instance. The parameter settings for each algorithm are determined through the Taguchi calibration widely utilized in published papers.

Table 7 provides the average 1-NHV values from 10 independent runs across 120 instances. For space reasons, only the results of the best four algorithms are presented here. Detailed results for all evaluation indicators are omitted due to space constraints, but are available upon request. As the smaller 1-NHV value indicates the better algorithm performance, the algorithms can be ranked in increasing order of the overall average 1-NHV values presented in Table 7. Namely, QMOHH ranks first as it achieves the best overall average value (0.198), demonstrating the superiority of QMOHH over all other compared algorithms. MOPSO ranks second with 0.222, SMPSO ranks third with 0.236, and MOEAD ranks fourth with 0.287. Among the methods, MOSA and MRSA exhibit the weakest performance.

This ranking clearly shows that QMOHH holds a clear advantage over the compared methods in terms of 1-NHV. In fact, the superior performance of QMOHH is attributed to adaptive operator selection with the Q-learning mechanism and the new density-aware leader selection strategy. These improvements together are in favor of QMOHH achieving a proper balance between exploration and exploitation, and hence QMOHH produces outstanding performance for the multi-objective RALSP.

To validate the performance differences statistically, this section conducts the non-parametric statistical test due to the violation of normality assumptions required by ANOVA. The Friedman test is conducted separately on the mean ranks of the algorithms for each performance indicator (GD, $I_\epsilon$, and 1-NHV). The results of the Friedman test reject the null hypothesis (p-value < 0.01) for all three indicators, confirming that there exist statistically significant differences among the algorithms' performances.

Fig 7 illustrates the means plot for the average ranks of the algorithms in terms of GD. Clearly, QMOHH achieves the lowest average rank, indicating its superior performance in convergence. Except for MOPSO, confidence intervals of

**Table 6.** Average 1-NHV values of QMOHH and its variants.

| Instance | MOHHv3 | MOHHv1 | MOHHv2 | MOHHv4 | QMOHH | Instance | MOHHv3 | MOHHv1 | MOHHv2 | MOHHv4 | QMOHH |
|---|---|---|---|---|---|---|---|---|---|---|---|
| 1 | 0.090 | 0.098 | 0.050 | 0.053 | 0.052 | 61 | 0.152 | 0.173 | 0.059 | 0.057 | 0.062 |
| 2 | 0.137 | 0.157 | 0.103 | 0.083 | 0.091 | 62 | 0.141 | 0.110 | 0.070 | 0.078 | 0.072 |
| 3 | 1.000 | 1.000 | 1.000 | 1.000 | 1.000 | 63 | 0.154 | 0.102 | 0.082 | 0.081 | 0.070 |
| 4 | 0.167 | 0.166 | 0.111 | 0.117 | 0.111 | 64 | 0.139 | 0.116 | 0.095 | 0.092 | 0.095 |
| 5 | 1.000 | 1.000 | 1.000 | 1.000 | 1.000 | 65 | 0.222 | 0.230 | 0.160 | 0.155 | 0.144 |
| 6 | 0.275 | 0.196 | 0.224 | 0.225 | 0.209 | 66 | 0.633 | 0.481 | 0.573 | 0.438 | 0.397 |
| 7 | 0.218 | 0.238 | 0.155 | 0.159 | 0.172 | 67 | 0.304 | 0.323 | 0.233 | 0.211 | 0.216 |
| 8 | 0.137 | 0.146 | 0.109 | 0.106 | 0.097 | 68 | 0.356 | 0.324 | 0.303 | 0.287 | 0.266 |
| 9 | 0.225 | 0.222 | 0.190 | 0.173 | 0.143 | 69 | 0.258 | 0.225 | 0.209 | 0.195 | 0.183 |
| 10 | 0.242 | 0.281 | 0.211 | 0.187 | 0.174 | 70 | 0.200 | 0.173 | 0.149 | 0.165 | 0.145 |
| 11 | 1.000 | 1.000 | 1.000 | 1.000 | 1.000 | 71 | 0.288 | 0.195 | 0.189 | 0.196 | 0.176 |
| 12 | 0.143 | 0.185 | 0.090 | 0.098 | 0.096 | 72 | 0.177 | 0.160 | 0.130 | 0.127 | 0.115 |
| 13 | 0.211 | 0.218 | 0.145 | 0.159 | 0.142 | 73 | 0.244 | 0.184 | 0.195 | 0.196 | 0.198 |
| 14 | 0.168 | 0.145 | 0.123 | 0.124 | 0.126 | 74 | 0.297 | 0.299 | 0.237 | 0.252 | 0.244 |
| 15 | 0.239 | 0.194 | 0.167 | 0.171 | 0.165 | 75 | 0.159 | 0.130 | 0.116 | 0.120 | 0.111 |
| 16 | 0.286 | 0.228 | 0.218 | 0.215 | 0.203 | 76 | 0.368 | 0.326 | 0.298 | 0.307 | 0.287 |
| 17 | 0.288 | 0.258 | 0.273 | 0.253 | 0.234 | 77 | 0.262 | 0.162 | 0.209 | 0.219 | 0.186 |
| 18 | 0.220 | 0.235 | 0.171 | 0.173 | 0.171 | 78 | 0.246 | 0.258 | 0.171 | 0.177 | 0.167 |
| 19 | 0.263 | 0.223 | 0.204 | 0.205 | 0.208 | 79 | 0.256 | 0.257 | 0.200 | 0.197 | 0.190 |
| 20 | 0.309 | 0.249 | 0.234 | 0.267 | 0.226 | 80 | 0.435 | 0.400 | 0.353 | 0.373 | 0.317 |
| 21 | 0.122 | 0.125 | 0.069 | 0.064 | 0.061 | 81 | 0.314 | 0.316 | 0.214 | 0.183 | 0.176 |
| 22 | 0.175 | 0.172 | 0.113 | 0.085 | 0.113 | 82 | 1.000 | 1.000 | 1.000 | 1.000 | 1.000 |
| 23 | 0.220 | 0.248 | 0.180 | 0.177 | 0.150 | 83 | 0.185 | 0.256 | 0.227 | 0.129 | 0.088 |
| 24 | 0.155 | 0.148 | 0.115 | 0.102 | 0.102 | 84 | 1.000 | 1.000 | 1.000 | 1.000 | 1.000 |
| 25 | 0.224 | 0.251 | 0.165 | 0.154 | 0.130 | 85 | 0.203 | 0.210 | 0.159 | 0.123 | 0.126 |
| 26 | 0.215 | 0.234 | 0.151 | 0.132 | 0.144 | 86 | 0.242 | 0.243 | 0.175 | 0.160 | 0.161 |
| 27 | 0.363 | 0.473 | 0.268 | 0.266 | 0.237 | 87 | 0.196 | 0.207 | 0.141 | 0.141 | 0.133 |
| 28 | 0.181 | 0.175 | 0.118 | 0.127 | 0.123 | 88 | 0.242 | 0.299 | 0.177 | 0.154 | 0.165 |
| 29 | 0.391 | 0.233 | 0.448 | 0.325 | 0.266 | 89 | 0.229 | 0.170 | 0.177 | 0.196 | 0.188 |
| 30 | 0.492 | 0.371 | 0.389 | 0.379 | 0.435 | 90 | 0.333 | 0.284 | 0.262 | 0.269 | 0.246 |
| 31 | 0.200 | 0.188 | 0.136 | 0.147 | 0.140 | 91 | 0.148 | 0.118 | 0.100 | 0.108 | 0.114 |
| 32 | 0.228 | 0.201 | 0.167 | 0.179 | 0.168 | 92 | 0.266 | 0.279 | 0.246 | 0.261 | 0.257 |
| 33 | 0.291 | 0.279 | 0.197 | 0.201 | 0.157 | 93 | 0.206 | 0.254 | 0.161 | 0.150 | 0.130 |
| 34 | 0.174 | 0.165 | 0.119 | 0.120 | 0.125 | 94 | 0.233 | 0.277 | 0.150 | 0.159 | 0.163 |
| 35 | 0.284 | 0.238 | 0.240 | 0.229 | 0.205 | 95 | 0.339 | 0.317 | 0.257 | 0.261 | 0.258 |
| 36 | 0.243 | 0.225 | 0.194 | 0.196 | 0.197 | 96 | 0.280 | 0.261 | 0.189 | 0.194 | 0.201 |
| 37 | 0.255 | 0.306 | 0.199 | 0.191 | 0.175 | 97 | 0.510 | 0.350 | 0.399 | 0.408 | 0.394 |
| 38 | 0.288 | 0.266 | 0.251 | 0.235 | 0.240 | 98 | 0.177 | 0.145 | 0.138 | 0.136 | 0.145 |
| 39 | 0.160 | 0.152 | 0.135 | 0.143 | 0.135 | 99 | 0.210 | 0.203 | 0.162 | 0.169 | 0.150 |
| 40 | 0.203 | 0.256 | 0.170 | 0.142 | 0.127 | 100 | 0.329 | 0.336 | 0.283 | 0.286 | 0.266 |
| 41 | 0.162 | 0.152 | 0.114 | 0.097 | 0.087 | 101 | 0.283 | 0.336 | 0.236 | 0.174 | 0.175 |
| 42 | 0.201 | 0.323 | 0.143 | 0.111 | 0.154 | 102 | 0.191 | 0.200 | 0.136 | 0.073 | 0.121 |
| 43 | 0.117 | 0.107 | 0.082 | 0.079 | 0.066 | 103 | 0.158 | 0.197 | 0.102 | 0.098 | 0.114 |
| 44 | 0.342 | 0.299 | 0.183 | 0.139 | 0.217 | 104 | 0.360 | 0.285 | 0.251 | 0.213 | 0.270 |
| 45 | 1.000 | 1.000 | 1.000 | 1.000 | 1.000 | 105 | 0.112 | 0.160 | 0.082 | 0.097 | 0.076 |
| 46 | 0.374 | 0.416 | 0.282 | 0.277 | 0.262 | 106 | 0.356 | 0.384 | 0.264 | 0.267 | 0.200 |

*(Continued)*

**Table 6.** (Continued)

| Instance | MOHHv3 | MOHHv1 | MOHHv2 | MOHHv4 | QMOHH | Instance | MOHHv3 | MOHHv1 | MOHHv2 | MOHHv4 | QMOHH |
|---|---|---|---|---|---|---|---|---|---|---|---|
| 47 | 0.431 | 0.369 | 0.323 | 0.316 | 0.341 | 107 | 0.234 | 0.241 | 0.175 | 0.161 | 0.151 |
| 48 | 0.350 | 0.302 | 0.272 | 0.261 | 0.247 | 108 | 0.315 | 0.285 | 0.235 | 0.232 | 0.213 |
| 49 | 0.204 | 0.230 | 0.145 | 0.173 | 0.139 | 109 | 0.283 | 0.177 | 0.171 | 0.171 | 0.169 |
| 50 | 0.518 | 0.418 | 0.456 | 0.475 | 0.433 | 110 | 0.175 | 0.181 | 0.124 | 0.122 | 0.117 |
| 51 | 0.178 | 0.145 | 0.137 | 0.149 | 0.135 | 111 | 0.534 | 0.452 | 0.477 | 0.462 | 0.448 |
| 52 | 0.535 | 0.538 | 0.464 | 0.435 | 0.404 | 112 | 0.457 | 0.367 | 0.341 | 0.334 | 0.289 |
| 53 | 0.232 | 0.234 | 0.181 | 0.173 | 0.164 | 113 | 0.267 | 0.289 | 0.242 | 0.235 | 0.188 |
| 54 | 0.217 | 0.171 | 0.163 | 0.169 | 0.144 | 114 | 0.389 | 0.424 | 0.342 | 0.341 | 0.315 |
| 55 | 0.533 | 0.402 | 0.426 | 0.431 | 0.438 | 115 | 0.368 | 0.441 | 0.343 | 0.358 | 0.320 |
| 56 | 0.194 | 0.146 | 0.156 | 0.165 | 0.154 | 116 | 0.267 | 0.229 | 0.203 | 0.197 | 0.193 |
| 57 | 0.244 | 0.245 | 0.246 | 0.223 | 0.199 | 117 | 0.384 | 0.498 | 0.275 | 0.319 | 0.296 |
| 58 | 0.392 | 0.285 | 0.300 | 0.283 | 0.273 | 118 | 0.430 | 0.376 | 0.309 | 0.320 | 0.276 |
| 59 | 0.533 | 0.351 | 0.366 | 0.391 | 0.338 | 119 | 0.281 | 0.245 | 0.216 | 0.230 | 0.205 |
| 60 | 0.404 | 0.395 | 0.344 | 0.383 | 0.353 | 120 | 0.231 | 0.234 | 0.179 | 0.185 | 0.161 |
|  |  |  |  |  |  | Avg | 0.305 | 0.290 | 0.248 | 0.242 | **0.232** |

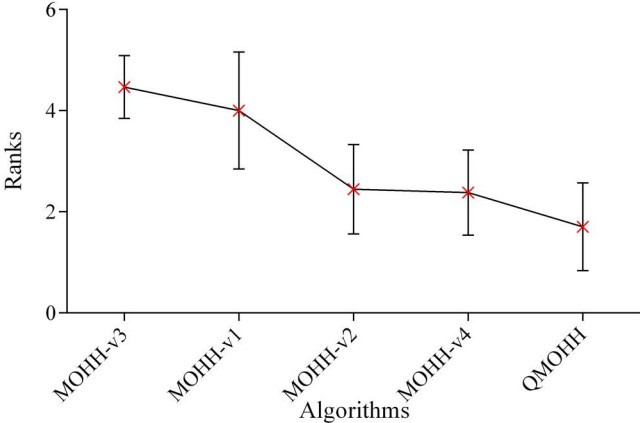

**Fig 6. Means plot of the average ranks and 95% confidence intervals of MOHH methods in terms of 1-NHV.**

QMOHH and the other eight algorithms do not overlap, and hence it is safe to say that QMOHH outperforms the other eight algorithms statistically in terms of GD. In short, QMOHH obtains the lowest average rank in terms of GD in this statistical analysis.

Similarly, Fig 8 presents the results for the $I_\epsilon$ indicator. QMOHH again attains the best (lowest) average rank. In this case, the confidence intervals of MOPSO, SMPSO and QMOHH overlap, which is common when multiple high-performing algorithms are compared. As the confidence intervals of QMOHH and the other seven algorithms do not overlap, QMOHH outperforms the other seven algorithms statistically in terms of $I_\epsilon$. In short, QMOHH consistently achieves the lowest mean rank and achieves competing performance again.

**Table 7. Results of the average 1-NHV by the four best-performing algorithms.**

| Instance | MOEAD | SMPSO | MOPSO | QMOHH | Instance | MOEAD | SMPSO | MOPSO | QMOHH |
|---|---|---|---|---|---|---|---|---|---|
| 1 | 0.061 | 0.067 | 0.060 | 0.031 | 61 | 0.089 | 0.072 | 0.047 | 0.032 |
| 2 | 0.085 | 0.090 | 0.072 | 0.063 | 62 | 0.061 | 0.055 | 0.032 | 0.030 |
| 3 | 1.000 | 1.000 | 1.000 | 1.000 | 63 | 0.122 | 0.135 | 0.091 | 0.065 |
| 4 | 0.162 | 0.171 | 0.127 | 0.106 | 64 | 0.102 | 0.096 | 0.080 | 0.100 |
| 5 | 1.000 | 1.000 | 1.000 | 1.000 | 65 | 0.190 | 0.201 | 0.160 | 0.104 |
| 6 | 0.174 | 0.154 | 0.118 | 0.136 | 66 | 0.346 | 0.226 | 0.180 | 0.193 |
| 7 | 0.202 | 0.201 | 0.168 | 0.139 | 67 | 0.225 | 0.195 | 0.185 | 0.121 |
| 8 | 0.156 | 0.140 | 0.130 | 0.095 | 68 | 0.261 | 0.212 | 0.195 | 0.179 |
| 9 | 0.224 | 0.209 | 0.180 | 0.117 | 69 | 0.241 | 0.182 | 0.183 | 0.145 |
| 10 | 0.207 | 0.202 | 0.167 | 0.130 | 70 | 0.174 | 0.158 | 0.145 | 0.102 |
| 11 | 1.000 | 1.000 | 1.000 | 1.000 | 71 | 0.160 | 0.152 | 0.129 | 0.101 |
| 12 | 0.156 | 0.136 | 0.116 | 0.058 | 72 | 0.195 | 0.162 | 0.135 | 0.093 |
| 13 | 0.186 | 0.150 | 0.140 | 0.095 | 73 | 0.258 | 0.218 | 0.207 | 0.199 |
| 14 | 0.130 | 0.123 | 0.108 | 0.092 | 74 | 0.297 | 0.220 | 0.185 | 0.154 |
| 15 | 0.240 | 0.182 | 0.175 | 0.154 | 75 | 0.172 | 0.107 | 0.125 | 0.105 |
| 16 | 0.367 | 0.233 | 0.232 | 0.199 | 76 | 0.249 | 0.130 | 0.168 | 0.164 |
| 17 | 0.270 | 0.213 | 0.189 | 0.188 | 77 | 0.250 | 0.148 | 0.212 | 0.172 |
| 18 | 0.236 | 0.215 | 0.194 | 0.140 | 78 | 0.180 | 0.138 | 0.094 | 0.091 |
| 19 | 0.329 | 0.246 | 0.211 | 0.167 | 79 | 0.249 | 0.174 | 0.148 | 0.093 |
| 20 | 0.271 | 0.190 | 0.170 | 0.168 | 80 | 0.387 | 0.356 | 0.322 | 0.309 |
| 21 | 0.057 | 0.061 | 0.042 | 0.022 | 81 | 0.196 | 0.162 | 0.108 | 0.087 |
| 22 | 0.124 | 0.103 | 0.075 | 0.064 | 82 | 1.000 | 1.000 | 1.000 | 1.000 |
| 23 | 0.114 | 0.058 | 0.051 | 0.071 | 83 | 1.000 | 1.000 | 1.000 | 1.000 |
| 24 | 0.153 | 0.131 | 0.092 | 0.066 | 84 | 1.000 | 1.000 | 1.000 | 1.000 |
| 25 | 0.180 | 0.154 | 0.136 | 0.072 | 85 | 0.115 | 0.089 | 0.069 | 0.070 |
| 26 | 0.243 | 0.258 | 0.190 | 0.122 | 86 | 0.133 | 0.130 | 0.084 | 0.079 |
| 27 | 0.314 | 0.269 | 0.254 | 0.143 | 87 | 0.328 | 0.229 | 0.147 | 0.143 |
| 28 | 0.231 | 0.139 | 0.135 | 0.121 | 88 | 0.219 | 0.127 | 0.108 | 0.078 |
| 29 | 0.252 | 0.148 | 0.145 | 0.142 | 89 | 0.257 | 0.196 | 0.167 | 0.205 |
| 30 | 0.375 | 0.205 | 0.239 | 0.263 | 90 | 0.248 | 0.167 | 0.153 | 0.174 |
| 31 | 0.256 | 0.218 | 0.167 | 0.123 | 91 | 0.175 | 0.112 | 0.091 | 0.090 |
| 32 | 0.344 | 0.314 | 0.241 | 0.192 | 92 | 0.239 | 0.204 | 0.199 | 0.178 |
| 33 | 0.254 | 0.231 | 0.178 | 0.109 | 93 | 0.270 | 0.261 | 0.210 | 0.102 |
| 34 | 0.186 | 0.137 | 0.142 | 0.098 | 94 | 0.221 | 0.197 | 0.168 | 0.104 |
| 35 | 0.313 | 0.249 | 0.269 | 0.245 | 95 | 0.258 | 0.243 | 0.196 | 0.191 |
| 36 | 0.233 | 0.182 | 0.188 | 0.173 | 96 | 0.250 | 0.195 | 0.176 | 0.152 |
| 37 | 0.261 | 0.226 | 0.196 | 0.134 | 97 | 0.353 | 0.271 | 0.237 | 0.283 |
| 38 | 0.242 | 0.149 | 0.130 | 0.128 | 98 | 0.200 | 0.153 | 0.123 | 0.122 |
| 39 | 0.166 | 0.145 | 0.150 | 0.121 | 99 | 0.231 | 0.206 | 0.173 | 0.139 |
| 40 | 0.240 | 0.213 | 0.206 | 0.102 | 100 | 0.295 | 0.248 | 0.262 | 0.211 |
| 41 | 0.101 | 0.081 | 0.048 | 0.042 | 101 | 0.265 | 0.266 | 0.135 | 0.110 |
| 42 | 0.086 | 0.078 | 0.070 | 0.052 | 102 | 0.059 | 0.054 | 0.038 | 0.036 |
| 43 | 0.120 | 0.090 | 0.093 | 0.070 | 103 | 0.118 | 0.080 | 0.091 | 0.081 |
| 44 | 0.193 | 0.075 | 0.121 | 0.114 | 104 | 0.211 | 0.118 | 0.119 | 0.144 |
| 45 | 1.000 | 1.000 | 1.000 | 1.000 | 105 | 1.000 | 1.000 | 1.000 | 1.000 |
| 46 | 0.335 | 0.298 | 0.218 | 0.170 | 106 | 0.223 | 0.227 | 0.195 | 0.101 |

*(Continued)*

**Table 7.** (Continued)

| Instance | MOEAD | SMPSO | MOPSO | QMOHH | Instance | MOEAD | SMPSO | MOPSO | QMOHH |
|---|---|---|---|---|---|---|---|---|---|
| 47 | 0.215 | 0.137 | 0.183 | 0.175 | 107 | 0.245 | 0.207 | 0.174 | 0.095 |
| 48 | 0.257 | 0.198 | 0.189 | 0.159 | 108 | 0.345 | 0.193 | 0.207 | 0.195 |
| 49 | 0.230 | 0.122 | 0.194 | 0.124 | 109 | 0.150 | 0.144 | 0.097 | 0.088 |
| 50 | 0.418 | 0.263 | 0.317 | 0.328 | 110 | 0.225 | 0.192 | 0.173 | 0.121 |
| 51 | 0.192 | 0.168 | 0.158 | 0.154 | 111 | 0.456 | 0.226 | 0.342 | 0.343 |
| 52 | 0.367 | 0.247 | 0.249 | 0.203 | 112 | 0.188 | 0.103 | 0.100 | 0.110 |
| 53 | 0.193 | 0.144 | 0.138 | 0.100 | 113 | 0.225 | 0.192 | 0.183 | 0.122 |
| 54 | 0.186 | 0.138 | 0.131 | 0.116 | 114 | 0.404 | 0.222 | 0.314 | 0.345 |
| 55 | 0.321 | 0.237 | 0.229 | 0.217 | 115 | 0.546 | 0.374 | 0.398 | 0.370 |
| 56 | 0.192 | 0.164 | 0.139 | 0.127 | 116 | 0.261 | 0.184 | 0.159 | 0.138 |
| 57 | 0.252 | 0.145 | 0.182 | 0.170 | 117 | 0.320 | 0.154 | 0.178 | 0.149 |
| 58 | 0.505 | 0.316 | 0.313 | 0.299 | 118 | 0.418 | 0.347 | 0.280 | 0.230 |
| 59 | 0.489 | 0.311 | 0.272 | 0.269 | 119 | 0.267 | 0.179 | 0.196 | 0.173 |
| 60 | 0.424 | 0.342 | 0.291 | 0.258 | 120 | 0.306 | 0.250 | 0.199 | 0.150 |
|  |  |  |  |  | Avg | 0.287 | 0.236 | 0.222 | **0.198** |

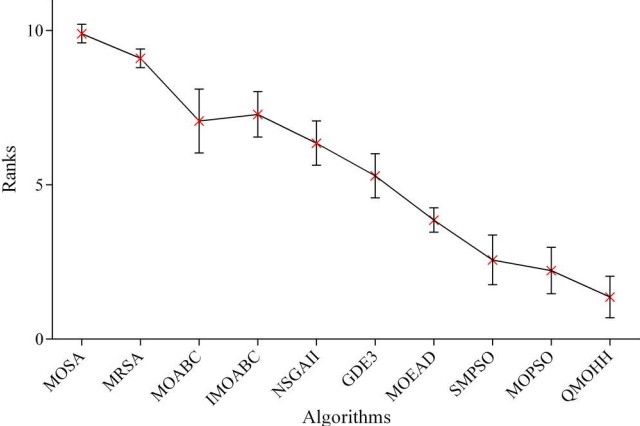

**Fig 7.  Means plot of the average ranks and 95% confidence intervals of the algorithms in terms of GD.**

Finally, Fig 9 illustrates the ranking results for the 1-NHV indicator. Consistent with the other evaluation indicators, QMOHH secures the top rank. The confidence intervals of QMOHH and MOPSO overlap, yet QMOHH's lower average rank signifies its superior ability to balance convergence and diversity in the obtained Pareto front.

To further validate pairwise differences, Dunn's multiple comparisons test is conducted as a post-hoc analysis following the Friedman test. The results show that QMOHH significantly outperforms all compared algorithms except MOPSO and SMPSO in terms of three evaluation indicators. These findings are consistent with the overlapping confidence intervals observed in Figs 7–9 and confirm that QMOHH achieves superior or competitive performance while being statistically superior to the majority of the state-of-the-art algorithms. The comparative study, along with the non-parametric statistical analysis, confirms the effectiveness and robustness of the proposed QMOHH in solving the multi-objective RALSP under

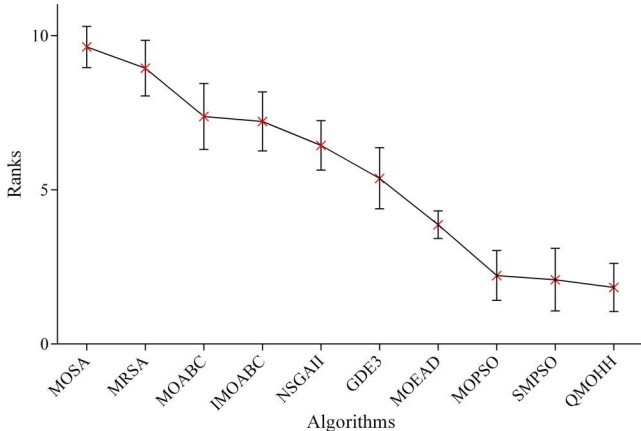

**Fig 8. Means plot of the average ranks and 95% confidence intervals of the algorithms in terms of $I_\epsilon$.**

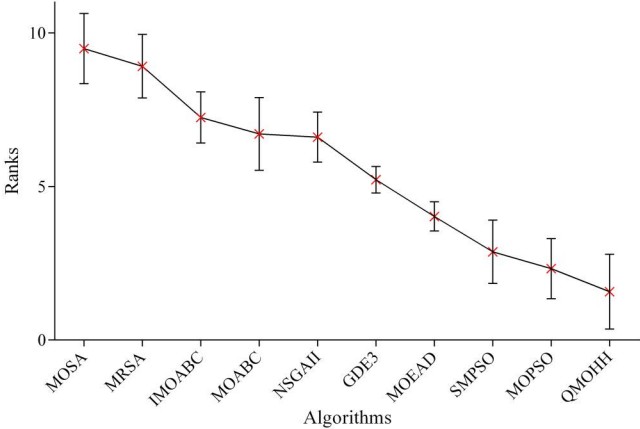

**Fig 9. Means plot of the average ranks and 95% confidence intervals of the algorithms in terms of 1-NHV.**

consideration. Despite some overlap in confidence intervals, QMOHH achieves the top ranking in terms of all three performance indicators; it could be concluded that the proposed methodology produces superior performance in solving the multi-objective RALSP.

## 7. Conclusions and future research directions

Faced with the increasing demand for customized and multi-variety production, reconfigurable assembly lines have gained prominence as a flexible manufacturing paradigm. This study tackles the multi-objective reconfigurable assembly line scheduling problem, which aims to minimize reconfiguration cost, production workload equalization, and logistics leveling simultaneously. A foundational contribution is the establishment of a new and linearized mathematical model comprising three MILP formulations, which enables exact problem characterization and provides a solid benchmark for algorithmic evaluation. Given the NP-hard nature of this problem, a novel Q-learning-based multi-objective hyper-heuristic algorithm is proposed. The algorithm operates within a unified search framework that dynamically selects and integrates multiple

metaheuristic operators—including particle swarm optimization, teaching–learning-based optimization, whale optimization algorithm, and grey wolf optimizer. A Q-learning mechanism is employed to adaptively choose the most promising operator at each search stage based on real-time performance feedback. The algorithm further incorporates a new density-aware leader selection strategy with a survival-time decay factor to select the global best solution for population evolution. This new density-aware leader selection strategy ensures an adaptive balance between exploration and exploitation by favoring superior solutions in sparse regions and increasing selection pressure on high-quality individuals in iterations.

A case study demonstrates that the models with the $\varepsilon$-constraint method can achieve a set of Pareto solutions, and the proposed hyper-heuristic method can effectively obtain a set of high-quality Pareto solutions, thereby validating the necessity and practicality of a multi-objective approach. Additionally, ablation experiments comparing four key variants of QMOHH confirmed that the Q-learning-driven adaptive operator selection, integration of diverse metaheuristic operators, and new density-aware leader selection strategy each contribute significantly to the algorithm's superior performance, outperforming simplified versions lacking these components. Extensive experiments on 120 generated benchmark instances further confirm the superiority of the proposed methodology. In comprehensive comparisons with nine state-of-the-art multi-objective algorithms—including multi-objective particle swarm optimization, non-dominated sorting genetic algorithm II, generalized differential evolution, multi-objective simulated annealing, and others—the proposed algorithm consistently achieved better performances in terms of several evaluation indicators. Statistical analysis further confirms the superiority of the proposed method in several evaluation indicators, demonstrating the effectiveness of the proposed Q-learning-driven operator selection and new density-aware leader selection strategy.

The proposed algorithm could be embedded into a production decision-support system to assist the line manager to achieve the proper product sequencing. The proposed model and QMOHH could obtain a set of high-quality Pareto solutions, and the line manager could select the most suitable scheduling scheme according to real-time production requirements. Hence, the developed model and QMOHH could enhance the operational efficiency and responsiveness of reconfigurable assembly lines.

Despite the promising results, several limitations are acknowledged. First, the proposed MILP models, although fully linearized and solvable by commercial solvers, are best suited for small-to-medium instances due to their exponential computational complexity. Second, the hyper-heuristic parameters are calibrated using the Taguchi method on a representative set of instances; while QMOHH performs robustly within the tested ranges, optimal parameter settings may vary for different problem structures or scales, and further tuning is required in new applications. Third, the benchmark instances, though generated based on realistic production scenarios, are synthetic in nature. And overfitting to these instances cannot be completely ruled out. Future work includes validation on additional real-world industrial cases to further assess the generalization capability of the proposed approach. Meanwhile, for future research, the proposed flexible hyper-heuristic framework could be extended to solve other complex scheduling problems, including mixed-model line balancing, integrated lot-streaming and scheduling in reconfigurable systems, and others. The algorithm could also be extended to dynamic disruptions or real-time data-driven scheduling to further enhance its practicality in industrial applications. And the simultaneous optimization of line balancing and product sequencing in reconfigurable environments is another promising research venue.

## Author contributions

**Conceptualization:** Zixiang Li.

**Data curation:** Guoliang Liu.

**Formal analysis:** Haoyi Zhao.

**Investigation:** Fan Chen.

**Methodology:** Zixiang Li.

**Supervision:** Xiangming Huang, Gaojie Lu.

**Validation:** Xiangming Huang.

**Visualization:** Gaojie Lu.

**Writing – original draft:** Haoyi Zhao.

**Writing – review & editing:** Fan Chen.

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
