## [Decision Letter · Decision Letter 0]

17 Mar 2026

PONE-D-26-07514Learning-based multi-objective hyper-heuristic algorithm for reconfigurable assembly line scheduling problemsPLOS One

Dear Dr. Li,

Thank you for submitting your manuscript to PLOS ONE. After careful consideration, we feel that it has merit but does not fully meet PLOS ONE’s publication criteria as it currently stands. Therefore, we invite you to submit a revised version of the manuscript that addresses the points raised during the review process.

Be sure to address these main issues in the revised version:

**Clarify the novelty** by explicitly distinguishing the proposed approach from existing Q-learning and hyper-heuristic frameworks and prior formulations (e.g., Yuan and Deng, 2017).**Improve reproducibility and transparency** by providing publicly accessible data/code, full instance-generation details, solver settings, parameters, and random seeds.**Strengthen validation and experimental analysis** through statistical significance testing, clearer demonstrations of model linearization, and an explanation of the ε-constraint implementation.**Enhance clarity and presentation** by standardizing notation, removing unnecessary formatting, and providing clearer intuition for key algorithmic components such as density-aware leader selection.

We look forward to receiving your revised manuscript.

Kind regards,

Babak Aslani, Ph.D.

Academic Editor

PLOS One

Reviewers' comments:

Reviewer's Responses to Questions

**Comments to the Author**

1. Is the manuscript technically sound, and do the data support the conclusions?

Reviewer #1: Yes

Reviewer #2: Yes

2. Has the statistical analysis been performed appropriately and rigorously? 

Reviewer #1: No

Reviewer #2: Yes

3. Have the authors made all data underlying the findings in their manuscript fully available?

Reviewer #1: Yes

Reviewer #2: No

4. Is the manuscript presented in an intelligible fashion and written in standard English?

Reviewer #1: Yes

Reviewer #2: Yes

5. Review Comments to the Author

Reviewer #1: My comments go thus:

1. Comment 1: The manuscript claims to propose a novel Q-learning-based multi-objective hyper-heuristic algorithm that integrates several metaheuristic operators (PSO, TLBO, WOA, and GWO). However, the novelty relative to existing reinforcement-learning-assisted metaheuristic frameworks is not sufficiently articulated. Many previous studies have already combined Q-learning with evolutionary or swarm-based optimization for adaptive operator selection. The authors should clearly explain what differentiates their approach from these existing methods, particularly regarding the structure of the hyper-heuristic, the operator adaptation mechanism, and the proposed density-aware leader selection strategy. A more explicit comparison with recent reinforcement-learning-assisted optimization frameworks would help establish the true methodological contribution of the study.

2. Comment 2: The experimental study evaluates the proposed algorithm using 100 generated benchmark instances, yet the manuscript does not clearly explain how these instances were generated. Key aspects such as parameter distributions, ranges for processing times, demand levels, reconfiguration costs, and part-frequency constraints are not described in sufficient detail. Without this information, the reproducibility of the experiments becomes limited. The authors should provide a clear description of the instance generation procedure, including parameter ranges, generation rules, and whether the datasets will be made publicly available.

3. Comment 3: Although the proposed algorithm is compared with nine state-of-the-art multi-objective algorithms, the manuscript does not provide sufficient statistical validation to support the claim that the proposed method consistently outperforms the competitors. The results are primarily reported through performance indicators without formal statistical testing. To strengthen the reliability of the conclusions, the authors should conduct appropriate statistical significance tests (e.g., Wilcoxon signed-rank test, Friedman test with post-hoc analysis) to verify whether the observed performance differences are statistically meaningful.

4. Comment 4: The proposed QMOHH algorithm relies on several parameters, including population size, learning rate, discount factor, ε-greedy exploration probability, and parameters specific to the underlying metaheuristic operators. However, the manuscript does not provide sufficient details about how these parameters were selected or tuned. It is unclear whether the parameter settings were chosen based on prior studies, preliminary experiments, or systematic tuning. Since algorithm performance can be highly sensitive to parameter settings, the authors should provide a detailed parameter table and explain the procedure used for parameter selection.

Reviewer #2: The manuscript examines the Reconfigurable Assembly Line Scheduling Problem (RALSP) by introducing three fully linearized multi objective MILP models designed to jointly minimize reconfiguration cost, production workload imbalances, and deviations in logistics leveling. Building on this modeling foundation, the authors propose a Q learning based multi objective hyper heuristic (QMOHH) that combines operators originating from PSO, TLBO, WOA, and GWO within a single adaptive search framework. The performance of this approach is tested on a set of 100 synthetically generated instances, with results compared against nine established state of the art multi objective algorithms. Based on these experiments, the authors report strong empirical performance and claim statistical superiority of the proposed QMOHH over the competing methods.

The manuscript benefits from several notable strengths. It offers an extensive and up to date literature review that covers foundational and contemporary research on reconfigurable manufacturing systems through 2025. The proposed hyper heuristic framework is well designed, modular, and readily extensible to other optimization settings. In addition, the case study is both realistic and well-motivated, effectively demonstrating the practical relevance of the proposed approach.

Originality and Contribution

The manuscript presents a series of three fully linearized MILP sub models whose structure and formulation are new, successfully overcoming limitations found in earlier nonlinear or logically inconsistent approaches. The integration of Q learning with several distinct metaheuristic operators into a unified hyper heuristic framework for the RALSP also represents an original and compelling methodological contribution. In addition, the numerical experiments are extensive and systematically performed, offering solid empirical support for the proposed approach.

Comments:

However, certain aspects of the contribution would benefit from clearer justification. In particular, the originality of the mathematical models should be demonstrated more explicitly, ideally through direct comparison with previously flawed formulations and a small illustrative example showing the specific ways in which earlier models fail. Some claims—such as the ability to produce “exact benchmarks”—should also be presented more cautiously, considering that exact solvability is inherently limited by instance size and computational scalability.

Overall, the work meets the originality expectations of PLOS ONE, requiring only minor clarifications to strengthen the presentation of its contributions.

Technical Quality of Experiments, Statistics, and Analyses — Revised Text

The technical component of the study shows several commendable aspects. The experimental design is broad and systematic, employing a set of one hundred benchmark instances evaluated over ten independent runs, which provides a solid statistical base for assessing the performance of the proposed method. The authors also make appropriate use of Friedman tests, a non parametric statistical tool well suited for comparing algorithms when data do not follow a normal distribution. In addition, the MILP formulations are presented with clarity, including precise definitions of variables, indices, and the linearization strategies required to make the models solvable by standard optimization tools.

Comments:

Despite these strengths, the manuscript raises several concerns that require attention.

1. Reproducibility is limited by the absence of publicly accessible code; although the Taguchi method is mentioned for parameter calibration, the documentation of this process is insufficient to allow independent replication.

2. Although the MILP models are described as tractable, the manuscript provides no computational evidence supporting this claim, such as solver times, memory usage, or scalability observations for larger instances.

3. The Q learning mechanism employs a discretization of optimization progress into ten states, but this choice appears arbitrary and is not justified theoretically or empirically.

Validity and Support of Conclusions — Revised Text

The conclusions presented in the manuscript are largely supported by the experimental evidence. The results consistently indicate that QMOHH delivers improved performance across the examined benchmark instances, and the statistical analyses reinforce the authors’ claim of superiority over competing algorithms. These findings demonstrate that the proposed method is effective and competitive within the context of the study.

Comments:

At the same time, several elements call for a more cautious interpretation.

1. Although the conclusion that QMOHH achieves superior performance is generally substantiated, the manuscript does not address the role of random seeds in shaping the results, leaving unanswered questions regarding the robustness and stability of the reported improvements.

2. Additionally, some performance indicators—such as the 1 NHV values for particular instances—display overlapping confidence intervals between QMOHH and other high-performing algorithms. This overlap suggests that the superiority of QMOHH is not uniform across all cases and therefore should not be stated in absolute terms.

Clarity, English Quality, and Organization — Revised Text

The manuscript is, overall, clearly written and logically organized, which makes it accessible and easy to follow. Its structure is coherent, the progression of ideas is generally smooth, and the included figures and tables provide substantial detail that supports the technical explanations.

Comments:

1. The manuscript would benefit from further refinement in several areas. Certain sections are noticeably lengthy and repeat ideas already stated elsewhere, suggesting that a more concise presentation would enhance clarity.

• Section 2 — Literature Review

This section is overly long and repeats core ideas about RMS and RALSP, as well as lengthy lists of studies that add little additional insight. Streamlining the discussion would improve clarity and highlight the paper’s own contribution more effectively.

• Section 3 — Problem Description and Model Formulations

The presentation of the MILP models is excessively detailed, reiterating explanations of variables, constraints, and linearization techniques multiple times. Reducing redundancy would make the section more focused and easier to follow.

• Section 6 — Experimental Results

The section is overloaded with repeated explanations of performance indicators and detailed descriptions of instance generation. Condensing the text would help foreground the key findings without compromising clarity.

2. Some visual elements, particularly Figure 5, appear overly complex and could be simplified so that their key messages become more immediately understandable.

In conclusion, several issues must be addressed:

1. The data availability statement should comply with PLOS policies, which require that all data, code, and instance generators be made publicly accessible.

2. The reproducibility of the study needs improvement by providing full parameter settings, initial random seeds, detailed solver configurations, runtime statistics, and the corresponding pseudocode or source code.

3. The validation of the mathematical models would benefit from a clearer demonstration of the correctness of the linearization—ideally through a small illustrative numerical example—and from a more explicit explanation of how the ε constraint procedure is implemented.

4. The manuscript should clarify its novelty by more directly contrasting the proposed models with previous formulations, such as that of Yuan and Deng (2017), and by including a comparative table showing how earlier nonlinear constraints differ from the newly proposed linear ones.

5. A more explicit discussion of limitations is needed, particularly regarding the scalability of the MILP models, the sensitivity of the hyper heuristic’s parameters, and the possibility of overfitting to the synthetic benchmark instances.

6. The notation must be made consistent throughout the text, for example by standardizing expressions such as nm versus nm.

7. Unnecessary bold formatting applied to variables and operators should be removed to maintain a clean mathematical style.

8. The manuscript would benefit from offering clearer intuition behind the density‑aware leader selection strategy to help readers better understand its role within the proposed algorithm.

6. PLOS authors have the option to publish the peer review history of their article (what does this mean?). If published, this will include your full peer review and any attached files.

Reviewer #1: No

Reviewer #2: No

---

## [Author Response · Author response to Decision Letter 1]

2 Apr 2026

Response to Reviewers

Journal: PLOS ONE

Submission ID: PONE-D-26-07514

Submission Title: Learning-based multi-objective hyper-heuristic algorithm for reconfigurable assembly line scheduling problems

Editor-in-Chief

Thank you for submitting your manuscript to PLOS ONE. After careful consideration, we feel that it has merit but does not fully meet PLOS ONE’s publication criteria as it currently stands. Therefore, we invite you to submit a revised version of the manuscript that addresses the points raised during the review process.

Response:

Dear Editor and Reviewers,

We appreciate the time and effort you have invested in helping us improve both the quality and organization of our paper. Each comment raised has been addressed and revisions have been made carefully. Revisions are given in 'blue' colored text in the revised manuscript for easy tracking. Please find below our point-by-point responses to your comments.

Kind regards

Corresponding author

Academic Editor

Comment 1-Academic Editor:

Clarify the novelty by explicitly distinguishing the proposed approach from existing Q-learning and hyper-heuristic frameworks and prior formulations (e.g., Yuan and Deng, 2017).

Response 1-Academic Editor:

We thank the editor for this suggestion. In the revised manuscript, the novelty of the proposed approach relative to existing Q-learning/hyper-heuristic frameworks and prior formulations (e.g., Yuan and Deng, 2017) has been explicitly clarified in the following locations:

1) Section 2.2 (Research gap and contributions): A direct methodological comparison is added, highlighting three key distinctions from existing reinforcement-learning-assisted metaheuristics.

2) Section 3.3 (Analysis of the formulated models): The proposed linearized MILP models are explicitly compared with the formulation of Yuan and Deng (2017), identifying the limitations of the prior work and the linearization strategies employed in this study.

3) Section 4.4 (Q-Learning mechanism for operator selection): The problem-independent state representation and the multi-objective reward function are further elaborated.

4) Section 4.5 (New density-aware leader selection strategy): The survival-time decay factor is explained as a novel mechanism to balance exploration and exploitation.

These revisions collectively distinguish our approach from existing methods.

Comment 2-Academic Editor:

Improve reproducibility and transparency by providing publicly accessible data/code, full instance-generation details, solver settings, parameters, and random seeds.

Response 2-Academic Editor:

We thank the editor for this suggestion. In the revised manuscript, we have significantly improved reproducibility and transparency. The following information is now publicly available or clearly described:

1) Publicly accessible data/code: All data, instance generation script, and source code of all algorithms are provided at https://github.com/zixiangliwust/Instances_RALSP under the MIT license (see Data Availability section).

2) Instance generation details: Full generation procedure (product types, demand patterns, station counts, processing times, reconfiguration costs, part requirements, part-frequency constraints, etc.) is described in Section 6.1 (Experimental design). The generation script uses a fixed random seed (42) to ensure determinism.

3) Solver settings: The MILP models are solved using IBM ILOG CPLEX 22.1.1 with a time limit of 3600 seconds per model and a MIP gap tolerance of 0.00 (optimality required), as stated in Section 3.2 and Section 5.

4) Parameter settings: Final calibrated parameters for QMOHH (population size, learning rate, discount factor, ε-greedy exploration, survival-time decay factor, mutation probabilities, termination criterion) are listed in Section 6.1. Parameters for all compared algorithms are also provided in the source code.

5) Random seeds: For instance generation, a fixed seed (42) is used. For algorithm runs, the random seed is set to the current system time for each independent run to ensure statistical independence (see Section 6.1).

These revisions fully address the reproducibility and transparency concerns.

Comment 3-Academic Editor:

Strengthen validation and experimental analysis through statistical significance testing, clearer demonstrations of model linearization, and an explanation of the ε-constraint implementation.

Response 3-Academic Editor:

We thank the editor for the suggestion. In the revised manuscript, we have strengthened validation and experimental analysis as follows:

1) Statistical testing: Friedman test and Dunn’s post-hoc analysis are reported in Section 6.4 (with visual results in Figures 7–9), confirming significant performance differences.

2) Model linearization: A clear demonstration of our three linearization strategies (auxiliary variables, big-M, two-stage ε-constraint) is provided in Section 3.3, explicitly contrasting with the prior formulation of Yuan and Deng (2017).

3) ε-constraint implementation: A step-by-step explanation (range determination, grid generation, two-stage handling of the bilinear term, and solution filtering) is given in Section 5.

These revisions fully address the comment.

Comment 4-Academic Editor:

Enhance clarity and presentation by standardizing notation, removing unnecessary formatting, and providing clearer intuition for key algorithmic components such as density-aware leader selection.

Response 4-Academic Editor:

We thank the editor for this suggestion. In the revised manuscript, we have enhanced clarity and presentation as follows:

1) Standardizing notation: All symbols (e.g., nm, nk, nr, nd) have been made consistent throughout the text, equations, and tables (see Section 3.2 and throughout).

2) Removing unnecessary formatting: Unnecessary bold formatting on variables and operators has been removed; only section headings and table headers retain bold styling (see Sections 3–6).

3) Clearer intuition for density aware leader selection: In Section 4.5, we added an intuitive explanation of the survival time decay factor as a “freshness” penalty, along with additional descriptions to clarify its role in balancing exploration and exploitation.

These revisions are highlighted in blue in the manuscript. We believe the clarity and presentation have been significantly improved.

Journal Requirements

Comment 1-Journal Requirements:

Response 1-Journal Requirements:

We have carefully formatted the manuscript according to PLOS ONE’s style requirements, including file naming and template guidelines. The revised submission complies with these standards.

Comment 2-Journal Requirements:

Please note that PLOS One has specific guidelines on code sharing for submissions in which author-generated code underpins the findings in the manuscript. In these cases, we expect all author-generated code to be made available without restrictions upon publication of the work. Please review our guidelines at https://journals.plos.org/plosone/s/materials-and-software-sharing#loc-sharing-code and ensure that your code is shared in a way that follows best practice and facilitates reproducibility and reuse.

Response 2-Journal Requirements:

All author-generated code has been made available without restrictions at https://github.com/zixiangliwust/Instances_RALSP under the MIT license, following PLOS ONE’s best practice guidelines.

Comment 3-Journal Requirements:

When completing the data availability statement of the submission form, you indicated that you will make your data available on acceptance. We strongly recommend all authors decide on a data sharing plan before acceptance, as the process can be lengthy and hold up publication timelines. Please note that, though access restrictions are acceptable now, your entire data will need to be made freely accessible if your manuscript is accepted for publication. This policy applies to all data except where public deposition would breach compliance with the protocol approved by your research ethics board. If you are unable to adhere to our open data policy, please kindly revise your statement to explain your reasoning and we will seek the editor's input on an exemption. Please be assured that, once you have provided your new statement, the assessment of your exemption will not hold up the peer review process.

Response 3-Journal Requirements:

We confirm that all data underlying the findings are now fully accessible at https://github.com/zixiangliwust/Instances_RALSP. The Data Availability Statement in the manuscript has been updated accordingly.

Comment 4-Journal Requirements:

Response 4-Journal Requirements:

We have reviewed the reviewer comments and found no recommendation to cite specific previously published works beyond those already included in the literature. No additional citations have been added.

Reviewer #1

Comment 1-overall:

My comments go thus

Response 1-overall:

We sincerely thank the reviewer for the thoughtful comments and overall assessment of our work. We have carefully addressed each of the specific points raised in the following responses, and the corresponding revisions have been made in the manuscript (highlighted in blue).

Comment 1-1:

The manuscript claims to propose a novel Q-learning-based multi-objective hyper-heuristic algorithm that integrates several metaheuristic operators (PSO, TLBO, WOA, and GWO). However, the novelty relative to existing reinforcement-learning-assisted metaheuristic frameworks is not sufficiently articulated. Many previous studies have already combined Q-learning with evolutionary or swarm-based optimization for adaptive operator selection. The authors should clearly explain what differentiates their approach from these existing methods, particularly regarding the structure of the hyper-heuristic, the operator adaptation mechanism, and the proposed density-aware leader selection strategy. A more explicit comparison with recent reinforcement-learning-assisted optimization frameworks would help establish the true methodological contribution of the study.

Response 1-1:

We thank the reviewer for this constructive comment. In the revised manuscript, we have substantially clarified the novelty of our QMOHH by explicitly contrasting it with existing reinforcement-learning-assisted metaheuristic frameworks. The following three key aspects, now clearly articulated in Section 2.2 (Research gap and contributions) of the revised manuscript, distinguish our approach from prior work.

1) Hyper-heuristic structure vs. single-algorithm embedding.

Most existing studies (e.g., Mosadegh et al., 2020; Zhang et al., 2024) embed Q-learning into a single metaheuristic (e.g., simulated annealing or an evolutionary algorithm) to tune parameters or select local search operators. In contrast, QMOHH acts as a hyper-heuristic that dynamically selects among eight complete operators derived from four distinct metaheuristics (PSO, TLBO, WOA, GWO). This design allows the framework to leverage the complementary strengths of multiple search paradigms within a unified process, rather than fine-tuning a single algorithm. (See Section 2.2, point 1.)

2) Problem-independent state representation and multi-objective reward.

Existing Q-learning frameworks often define the state using algorithm-specific metrics (e.g., population diversity, temperature) and reward based on scalar fitness improvement. By contrast, QMOHH defines the state purely by search progress (discretized into ten intervals) – a problem-independent design that requires no instance-specific tuning. Moreover, the reward is defined as the number of newly added non-dominated solutions to the Pareto archive, directly aligning the Q-learning objective with the goal of multi-objective optimization (expanding the Pareto front) rather than relying on single-objective proxies. (See Section 2.2, point 2, and Section 4.4.)

3) Density-aware leader selection with survival-time decay.

Conventional multi-objective algorithms select leaders based on crowding distance or random selection from the Pareto front. QMOHH introduces a survival-time decay factor γ^(τ_i ) that progressively reduces the selection attractiveness of long-surviving solutions. This mechanism prevents premature convergence and promotes sustained exploration in sparse regions. To the best of our knowledge, this is the first history-aware density mechanism in multi-objective hyper-heuristics. (See Section 2.2, point 3, and Section 4.5.)

In addition, the revised manuscript now includes a direct methodological comparison with recent reinforcement-learning-assisted optimization frameworks (e.g., Mosadegh et al., 2020; Zhang et al., 2024; Meng et al., 2025) within Section 2.2. We believe these clarifications, together with the ablation study in Section 6.3 (where replacing Q-learning with random selection or DQN leads to significantly worse performance), convincingly establish the novelty and effectiveness of the proposed QMOHH.

Comment 1-2:

The experimental study evaluates the proposed algorithm using 100 generated benchmark instances, yet the manuscript does not clearly explain how these instances were generated. Key aspects such as parameter distributions, ranges for processing times, demand levels, reconfiguration costs, and part-frequency constraints are not described in sufficient detail. Without this information, the reproducibility of the experiments becomes limited. The authors should provide a clear description of the instance generation procedure, including parameter ranges, generation rules, and whether the datasets will be made publicly available.

Response 1-2:

We thank the reviewer for emphasizing the importance of reproducibility. To address this comment, we have taken the following actions:

1) Instance generation details: In Section 6.1 of the revised manuscript, we now provide a comprehensive description of the instance generation procedure, including parameter distributions, ranges for processing times, demand levels, reconfiguration costs, and part-frequency constraints.

2) Deterministic generation: All random parameters are generated using a fixed random seed (42) in a dedicated random.Random instance. This ensures that running the instance generation script multiple times produces exactly the same 120 instance files. The seed and the entire generation logic are explicitly documented in the source code.

3) Public availability: The complete instance generation script (generate_instances.py), the generated 120 instance files, and the source code of all compared algorithms have been uploaded to a public GitHub repository under the MIT license: https://github.com/zixiangliwust/Instances_RALSP. The Data Availability statement in the manuscript has been updated accordingly.

4) Random seeds in algorithm runs: For the stochastic optimization algorithms (QMOHH and the nine competitors), each independent run uses the current system time as the random seed to ensure statistical independence across the 10 runs per instance. The fixed seed for instance generation guarantees that all algorithms are tested on exactly the same input data, while the varying seeds for algorithm runs provide a fair basis for statistical comparisons.

These revisions are highlighted in blue in the revised manuscript (see Section 6.1 and Data Availability). We believe these changes fully address the reviewer's concern and greatly improve the transparency and reproducibility of our study.

Comment 1-3:

Although the proposed algorithm is compared with nine state-of-the-art multi-objective algorithms, the m

---

## [Decision Letter · Decision Letter 1]

23 Apr 2026

Learning-based multi-objective hyper-heuristic algorithm for reconfigurable assembly line scheduling problems

PONE-D-26-07514R1

Dear Dr. Li,

We’re pleased to inform you that your manuscript has been judged scientifically suitable for publication and will be formally accepted for publication once it meets all outstanding technical requirements.

Kind regards,

Babak Aslani

Academic Editor

PLOS One

Additional Editor Comments (optional):

Reviewers' comments:

Reviewer's Responses to Questions

**Comments to the Author**

1. If the authors have adequately addressed your comments raised in a previous round of review and you feel that this manuscript is now acceptable for publication, you may indicate that here to bypass the “Comments to the Author” section, enter your conflict of interest statement in the “Confidential to Editor” section, and submit your "Accept" recommendation.

Reviewer #1: All comments have been addressed

Reviewer #2: All comments have been addressed

2. Is the manuscript technically sound, and do the data support the conclusions?

Reviewer #1: Yes

Reviewer #2: Yes

3. Has the statistical analysis been performed appropriately and rigorously? 

Reviewer #1: Yes

Reviewer #2: Yes

4. Have the authors made all data underlying the findings in their manuscript fully available?

Reviewer #1: Yes

Reviewer #2: Yes

5. Is the manuscript presented in an intelligible fashion and written in standard English?

Reviewer #1: Yes

Reviewer #2: Yes

6. Review Comments to the Author

Reviewer #1: After carefully reviewing the revised manuscript and the authors’ point-by-point responses, I am satisfied that the substantive concerns raised during the review process have been addressed adequately. The revision has improved the manuscript in terms of clarity, methodological justification, reproducibility, and presentation, and the authors have responded constructively to the comments from both the reviewers and the editor. In its current form, I consider the manuscript suitable for publication.

Reviewer #2: Thank you for the effort you put into this revision. I have carefully reviewed your responses and the updated manuscript. You have successfully addressed the comments and requests made during the initial review.

While some explanations in the response letter could have been a bit more exhaustive to fully reflect the quality of your work, the manuscript itself has improved significantly. In particular, the inclusion of the statistical analysis and the clarification of the density-aware leader selection strategy have made the paper much stronger.

7. PLOS authors have the option to publish the peer review history of their article (what does this mean?). If published, this will include your full peer review and any attached files.

Reviewer #1: No

Reviewer #2: **Yes:** Constantin Ilie, PhD

---

## [Editor Report · Acceptance letter]

PONE-D-26-07514R1

PLOS One

Dear Dr. Li,

I'm pleased to inform you that your manuscript has been deemed suitable for publication in PLOS One. Congratulations! Your manuscript is now being handed over to our production team.

Kind regards,

on behalf of

Dr. Babak Aslani

Academic Editor

PLOS One